# FoREST: Frame of Reference Evaluation in Spatial Reasoning Tasks

## Abstract

Spatial cognition is one fundamental aspect of human intelligence. A key factor in spatial cognition is understanding the frame of reference (FoR) that identifies the perspective of spatial relations. However, the AI research has paid very little attention to this concept. Specifically, there is a lack of dedicated benchmarks and in-depth experiments analyzing large language models' (LLMs) understanding of FoR. To address this issue, we introduce a new benchmark, **F**rame **o**f **R**eference **E**valuation in **S**patial Reasoning **T**asks (FoREST) to evaluate LLMs ability in understanding FoR. We evaluate the LLMs in identifying the FoR based on textual context and employ this concept in text-to-image generation. Our results reveal notable differences and biases in the FoR identification of various LLMs. Moreover, the bias in FoR interpretations impacts the LLMs' ability to generate layouts for text-to-image generation. To improve spatial comprehension of LLMs, we propose Spatial-Guided (SG) prompting, which guides the model in exploiting the types of spatial relations for a more accurate FoR identification. The SG prompting improves the overall performance of FoR identification by alleviating their bias towards specific frames of reference. Eventually, incorporating the FoR information generated by SG prompting in text-to-image leads to a more accurate visualization of the spatial configuration of objects.

## 1 Introduction

Spatial reasoning plays a significant role in human cognition and conducting daily activities. It is also a crucial aspect in many AI problems, including language grounding (Zhang & Kordjamshidi, 2022; Yang et al., 2024), navigation (Yamada et al., 2024), computer vision (Liu et al., 2023; Chen et al., 2024), medical domain (Gong et al., 2023), and image generation (Gokhale et al., 2023). One key concept in spatial cognition is the frame of reference (FoR), which identifies the perspective of spatial expressions. Levinson (2003) initially defines three basic FoR classes: intrinsic, relative, and absolute. The intrinsic FoR describes spatial expressions based on the viewer's perspective, while the relative FoR uses the object's perspective. The last type is the absolute FoR, which uses environmental cues such as cardinal directions. The framework from Tenbrink (2011), which is the main FoR framework of our work, expanded these basics. These FoR concepts have been studied extensively in cognitive linguistics (Edmonds-Wathen, 2012; Vukovic & Williams, 2015). Additionally, understanding this concept is significant for several AI applications. An important application is embodied AI. Particularly in a real environment, an instruction-giver and instruction-follower have different perspectives, and there are potential variations in their usage of FoRs. In such a setting, the model must comprehend the dynamic changes in the FoR (perspective changes) in the instruction to perform the task effectively. FoR comprehension can benefit other applications, such as video narrative generation and 3D scene construction based on text. The recent spatial evaluation benchmarks have paid less attention to the importance of FoRs. For instance, the textual-only benchmarks Shi et al. (2022); Mirzaee & Kordjamshidi (2022); Rizvi et al. (2024) concentrate on the complex reasoning task; however, they limit the evaluation to intrinsic FoR, using one object as the center of coordinates. Similarly, text-to-image benchmarks (Gokhale et al., 2023; Huang et al., 2023; Cho et al., 2023a;b) often assume a camera perspective for spatial expressions. This kind of bias in the datasets potentially restricts the situated spatial reasoning abilities in dynamic environments and interactive settings where the perspective can change.

To systematically investigate the concept of FoR in spatial understanding and provide new resources, we introduce **F**rame **o**f **R**eference **E**valuation in **S**patial Reasoning **T**asks (FoREST) benchmark to assess models' ability to understand FoR classes from textual descriptions and extend this to grounding and visualization. Our dataset consists of two splits: ambiguous (A-split) and clear (C-split). The A-split contains spatial expressions with FoR ambiguity, meaning multiple valid FoRs can apply to the explained situation. In contrast, the C-split has spatial expressions with only one valid FoR. This design allows us to evaluate models' understanding of spatial expressions in ambiguous and clear contexts. We conduct experiments with large language models (LLMs) to identify FoR classes in spatial expressions and employ this concept in text-to-image models. Our findings reveal performance differences across FoR classes and show that LLMs tend to be biased toward particular FoRs when spatial expressions with ambiguous FoRs are provided. The bias is also evident in diffusion models that use LLM-generated layouts in the image generation pipeline. These diffusion models tend to perform better in one specific FoR class. To improve spatial comprehension of LLMs, we propose Spatial-Guided (SG) prompting, which encourages models to consider the type of spatial relations, particularly directional, topological, and distance types of relations in their reasoning process for a more accurate FoR identification. Our results confirm that these relations provide essential information to help LLMs accurately identify FoR classes. In addition, we exploit the impact of FoR identification on downstream tasks like text-to-image generation. We show that FoR identification can enhance layout generation, ultimately benefiting text-to-image generation performance.

To summarize our contributions, 1. We introduce the FoREST benchmark to systematically evaluate large language models' abilities to identify FoR classes from textual spatial expressions, experimenting with various in-context learning approaches for FoR identification. 2. We assess the impact of using FoR information on text-to-image generation using diffusion models, including stable and layout diffusion models. 3. We propose a new prompting approach that considers the types of spatial relations in its reasoning process and improves FoR identification and image generation quality.

## 2 PRIMITIVES

We review three aspects of spatial information expressed in language: spatial roles, spatial relations, and frame of reference.

**Spatial Roles.** We use the main conceptual roles defined in spatial language literature (Kordjamshidi et al., 2010; Tenbrink, 2011) These roles include Locatum (L), Relatum (R), and Perspective. The **locatum** represents the object described in the spatial expression. While the **relatum** represents another object used to describe the location of the locatum. Lastly, **perspective** is defined as the origin of a coordinate system used as the basis for determining the direction. For example, "a cat is to the left of a dog from the owner." In this example, a cat is the locatum, a dog is a relatum, and the perspective is the owner's coordinate.

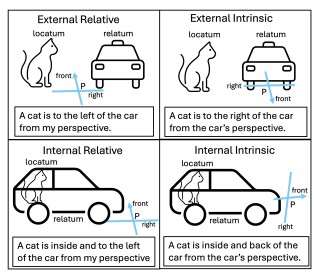

Figure 1: Illustration of FoR classes. The Cat is the locatum, the Car is the relatum, and the arrow indicates the perspective.

**Spatial Relations.** When dealing with spatial knowledge representation and reasoning, often three main relations categories are considered: directional, topological, and distance (Hernández, 1994; Cohn & Renz, 2008; Kordjamshidi et al., 2010).

1. Directional: These relations define one object's direction from another based on specific coordinates. Examples of relations include left, right, above, and below.

2. Topological: These relations describe the containment between two objects, such as inside.

3. Distance: These relations provide qualitative and quantitative relations between entities. Examples of qualitative distance relations are near and far, and quantitative distance relations are 3km.

**Spatial Frame of Reference.** We use the four frames of reference investigated in-depth in the cognitive linguistic studies (Tenbrink, 2011) and are defined as follows.

1. *external intrinsic*. It describes a spatial relation based on the relatum's perspective, which does not contain the locatum. The top-right image in Figure 1 illustrates this scenario with the sentence, "A cat is to the right of the car from the car's perspective."

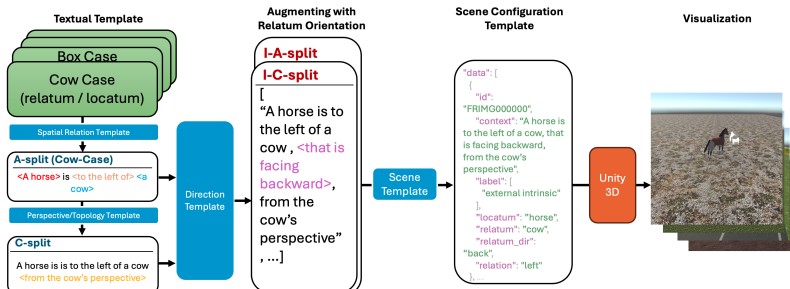

Figure 2: The pipeline of creating the FoREST dataset starts by selecting the locatum and relatum based on defined FoR cases. Next, a spatial template is applied to generate the A-split, which is then extended into the C-split by applying a topology/perspective template. Afterward, the I-C-split and I-A-split are created by including a direction template into the C-split and A-split. Finally, scene configurations are generated from the I-C-split and I-A-split to create visualizations using Unity3D.

2. *external relative*. It presents a spatial relation based on the observer's perspective, which may not presented in the context. The top-left image in Figure 1 shows an example with the sentence, "A cat is to the left of a car from my perspective."

3. *internal intrinsic*. It expresses a spatial relation based on the relatum's perspective, which contains the locatum. The bottom-right image in Figure 1 illustrates this circumstance with the sentence, "A cat is inside and back of the car from the car's perspective."

4. *internal relative*. It describes a spatial relation from the observer's perspective where the locatum is inside the relatum. The bottom-left image in Figure 1 displays this relation with the sentence, "A cat is inside and to the left of the car from my perspective."

# 3 FoREST Dataset Construction

We propose a new problem setting to identify the frame of reference (FoR) in linguistic expressions to evaluate the LLMs' understanding on spatial frames of reference(FoR). In this setting, the language model receives a textual spatial explanation as input, denoted as $T$, and the model outputs an FoR class in $FoR$={external intrinsic, external relative, internal relative, internal intrinsic} according to the primitives defined in Section 2. We introduce the **F**rame **of** **R**eference **E**valuation in **S**patial Reasoning **T**asks (FoREST) benchmark to evaluate models' performance on this problem. We should note that identifying FoR is challenging and, in some cases, inherently ambiguous. For example, in "a cat is to the left of a dog.", It has two correct interpretations. The first one is *external relative* FoR interpretation, "a cat is to the left of a dog from the camera's perspective." Another valid interpretation for *external intrinsic* FoR is "a cat is to the left of a dog from the dog's perspective." To distinguish clear from ambiguous cases, we create two splits for our FoREST dataset: ambiguous (A-split) and clear (C-split). Spatial expressions in the A-split can have more than one valid FoR, while C-split expressions only have one valid FoR.

## 3.1 FoR categories based on Relatum type

Using the FoR classes defined in Section 2, we found that two properties of relatum cause FoR ambiguity. The first property is the relatum's intrinsic direction. It creates ambiguity between intrinsic and relative FoR classes since spatial relations can originate from both the relatum's and observer's perspectives. The second is the relatum's affordance as a container. It introduces the ambiguity between internal and external FoR classes since spatial relations can refer to the inside and outside of the relatum. We use the combination of these two properties to define four cases of relatum: the cow case, box case, car case, and pen case. We use these cases to divide the A-split of our dataset into four subsets. Then, we create clear counterparts of these cases to generate the C-split of our dataset. There are two types of clear cases. The first type is inherently clear from the context, such as "a pencil is to the right of a pen." In this case, there are no different interpretations about the spatial configuration of the two objects. However, another type needs additional information to be clear, such as "A cat is to the left of the dog." In this type, we add a clause clarifying the perspective or

topology. For example, "the cat is to the left of the dog from the dog's perspective." In the following, we further clarify the four ambiguous cases based on the properties of the relatum.

**Case 1: Cow Case**. We create a cow case as a subset of our A-split. We select a relatum with intrinsic directions but without affordance as the container. The obvious example is a cow, which should not be a container but has a front and back. In such a case, the relatum potentially provides a perspective for spatial relations. Thus, the applicable FoR classes are $FoR = \{external\ intrinsic,\ external\ relative\}$. We explicitly augment such cases with perspective information to resolve the ambiguity and add their clear counterparts to the C-split. To specify the perspective, we use templates for augmenting clauses, such as "from {relatum}'s perspective" for *external intrinsic* or "from my perspective" for *external relative*. An example of A-split context is "a cat is to the right of the cow." The counterparts included in the C-split are "a cat is to the right of the cow from cow's perspective." for *external intrinsic* and "a cat is to the right of the cow from my perspective" for *external intrinsic*.

**Case 2: Box Case.** We create a box-case subset as part of the A-split. Unlike the cow case, the relatum selected in this subset can be a container but lacks intrinsic directions. For example, a box can serve as a container without having intrinsic directions. An internal FoR can be established since the relatum can be a container. Accordingly, the applicable FoR classes of this context are $FoR = \{external\ relative,\ internal\ relative\}$, causing the ambiguity. To include their unambiguous counterparts in the C-split, we explicitly specify the topology between locatum and relatum by adding "inside" for *internal relative* and "outside" for *external relative* in the spatial expression. An example of the A-split context is "A cat is to the right of the box." The counterpart for *internal relative* is "a cat is inside and to the right of the box." The counterpart for *external relative* is "a cat is outside and to the right of the box." We add both counterparts in the C-split.

**Case 3: Car Case.** We introduce the third case subset of A-split, Car case. We select the relatum with intrinsic direction and affordance as a container for this case. With these two properties, the relatum can provide the perspective for spatial relations and contain the locatum, allowing all FoR classes. An obvious example is a car that can be a container with intrinsic directions. Therefore, the applicable frames of reference classes are $FoR = \{external\ relative\ external\ intrinsic,\ internal\ intrinsic,\ internal\ relative\}$, which introduces FoR ambiguity. We resolve this ambiguity by including perspective and topology information to create clear counterparts for the C-split. The template for augment clauses is reused from the Cow case and Box case for perspective and topology information, respectively. A proper example of context in A-split is "a person is in front of the car." The four counterparts to include in the C-split are "a person is outside and in front of the car from the car itself" for *external intrinsic*, "a person is outside and in front of the car from the observer" for *external relative*, "a person is inside and in front of the car from the car itself" for *internal intrinsic*, and "a person is inside and in front of the car from the observer" for *internal relative*.

**Case 4: Pen Case.** We called the last subset of A-split with the Pen case. The last case covers the circumstance that the relatum neither has the intrinsic direction nor the affordance as a container. An obvious example is a pen that does not have a left or right direction nor the ability to be a container. Lacking these two properties, the created context should be clear and have one applicable FoR, $FoR = \{external\ relative\}$. There is no ambiguity to clarify since there is only one valid FoR class. Therefore, we can reuse it in the C-split without modifications. An example of such a context is "the book is to the left of a pen."

## 3.2 Context Visualization

As a part of the dataset, we include the image visualizations of spatial expressions. In intrinsic FoR classes, the relatum's perspective influences how we position the locatum when visualizing spatial expressions, leading to ambiguity in the position of objects in the scene. For example, given the expression "a cow is to the right of a car relative to the car," with the car's position fixed in the scene, the cows can be placed in different positions depending on the car's orientation. To address this issue, we extend the context in both splits of FoREST by adding the relatum's orientation information. To specify the relatum's orientation, we use templates such as "facing forward." For instance, "a cat is to the left of a dog" is extended to "a cat is to the left of a dog, facing forward." In this way, we obtain I-A-split from A-split and I-C-split from C-split. We restrict I-A-split and I-C-split to external FoR classes to avoid occlusion in the visualization since one object can become invisible in internal FoR classes. We then create scene configurations based on the spatial expressions in

I-A-split and I-C-split, as illustrated in Figure 2. We use the Unity-3D simulator [1] to process scene configurations and generate four visualizations for each one. The detail on the simulation is provided in the Appendix B.

## 3.3 RELATUM/LOCATUM SELECTION

We selected nine object sets to support the four FoR cases defined above. For instance, an example set of objects is "small objects with intrinsic direction." Selected objects in this group, such as dogs and cats, are guaranteed to have intrinsic direction without the affordance of being containers. This set is used to create the Cow Case context and visualization. All sets of objects are in the Appendix B. The total number of selected objects is 20, enough to cover all defined FoR cases.

## 3.4 DATASET CREATION PROCEDURE

The pipeline is illustrated in Figure 2 to combine all the above-explained procedures. First, we select a set of locatum and relatum based on the FoR cases defined in Section 3.1 to form A-split spatial expressions. We substitute the actual locatum and relatum objects in the Spatial Relation template, "<locatum> <spatial relation> <relatum>." In the figure, left is the spatial relation, locatum is a horse, and relatum is a cow. After obtaining the A-split contexts, we create their counterparts using the perspective/topology clauses described in Section 3.1 represented in yellow text. Next, we apply the orientation template described in Section 3.2 to prepare the context for the visualization. We then create the scene configuration from modified spatial expression and send it to the simulator to finalize visualizations. The dataset statistic is in Appendix A, and the complete sets of all patterns and entities are included in Appendix B.

# 4 MODELS AND TASKS

## 4.1 FoR IDENTIFICATION

**Task.** We evaluate the LLMs' performance in recognizing the FoR classes from given spatial expressions. The LLMs receive aspatial expression, denoted as $T$, and output one FoR class, $F$, from the valid set of FoR classes, $F \in FoR = \{$external relative, external intrinsic, internal intrinsic, internal relative$\}$. All in-context learning examples are in the Appendix C.

**Zero-shot model.** We follow the regular setting of *zero-shot* prompting. We only provide instruction to LLM with spatial context. The instruction prompt briefly explains each class of the FoR and candidate answers for the LLM. We called the LLM with the instruction prompt and $T$ to find $F$.

**Few-shot model.** We manually craft four spatial expressions for each FoR class. To avoid creating bias, each spatial expression is ensured to fit in only one FoR class. These expressions serve as examples of our *few-shot*setting. We provide these examples in addition to the instruction as a part of the prompt, followed by $T$ and query $F$ from the LLM.

**Chain-of-Thought (CoT) model.** To create CoT (Wei et al., 2023) examples, we modify the prompt to require reasoning before answering. Then, we manually crafted reasoning explanations with the necessary information for each example used in few-shot. Finally, we call the LLMs, adding modified instructions to updated examples, followed by $T$ and query $F$.

**Spatial-Guided Prompting (SG) model.** We hypothesize that the general spatial relation types defined in Section 2 can provide meaningful information for recognizing FoR classes. For instance, a topological relation, such as "inside," is intuitively associated with an internal FoR. Therefore, we propose Spatial-Guided Prompting to direct the model in identifying the type of relations before querying $F$. We revise the prompting instruction to guide the model in considering these three aspects. Then, we manually explain these three aspects. We specify the relation's origin from the context for direction relations, such as "the left direction is relative to the observer." We hypothesize that this information helps the model distinguish between intrinsic and relative FoR. Next, we specify whether the locatum is inside or outside the relatum for topological relations. This information should help distinguish between internal and external FoR classes. Lastly, we provide the potential

---
[1]https://unity.com

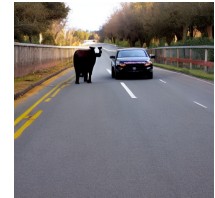

(a) An image generated from SD-2.1.  (b) An image generated from Llam3-8B + GLIGEN.

Figure 3: Two images generated from the ambiguous spatial expression "A car is to the right of a cow." (a) is correct by intrinsic FoR interpretation, while (b) is correct by relative FoR interpretation. These images only show the examples of possible interpretations of spatial expression in A-split that can be interpreted using multiple FoR classes.

quantitative distance, e.g., far. This quantitative distance further encourages identifying the correct topological and directional relations. Eventually, we insert these new explanations in examples and call the model with the updated instructions followed by $T$ to query $F$.

### 4.2 TEXT-TO-IMAGE (T2I)

**Task.** The input to the text-to-image is a spatial expression, $T$, and output from the model is a generated image, denoted as $I$, corresponding to given $T$. This task aims to determine the diffusion models' ability to consider FoR by assessing their generated images.

**Stable Diffusion models.** We evaluate the performance of the stable diffusion models for the simplest baseline of T2I models. This model only needs the scene description as input. Therefore, we provide $T$ to the model and expect an output image of $I$.

**Layout Diffusion models.** We evaluate the Layout Diffusion model for more advanced T2I models. The layout diffusion model has two phases: text-to-layout and layout-to-image. As the LLMs can be used to generate the bounding box layout (Cho et al., 2023b; Lian et al., 2024), we provide $T$ to LLMs with the instruction to generate the layout including bounding box coordinates for each object in the format of {object: $[x, y, w, h]$}, where $x$ and $y$ represent the starting point of the bounding box and $h$ and $w$ represent the height and width of the bounding box. After generating the bounding box coordinates, they are provided with $T$ as an additional input for the layout-to-image model to create the output image, $I$.

**Spatial-Guide Layout Diffusion models.** We propose Spatial-Guide Layout Diffusion pipeline for image generation, which introduces an additional step before the text-to-layout phase. This step involves obtaining the FoR information from $T$, denoted as $S(T)$. We guide LLMs to extract direction, topology, and distance information from $T$ to generate $S(T)$. Following the SG prompting procedure, we create examples for this step. Then, we provide examples to help the model understand the task and generate $S(T)$. Once $S(T)$ is generated, it is used as supplementary information to guide the LLMs in generating bounding box coordinates. This model allows us to consider FoRs in image generation and assess their impact on the T2I task. After obtaining the bounding box coordinates, we follow the same outline in Layout Diffusion to generate the final image.

## 5 EXPERIMENTAL RESULTS

### 5.1 EVALUATION METRICS

**FoR Identification.** We report the accuracy of the model on the multi-class classification task. Note that the expressions in A-split can have multiple correct answers. Therefore, we consider the prediction correct when it is in one of the valid FoR classes for the given spatial expression.

**T2I.** To evaluate the generated images, we assess the generated objects and their spatial relationships. To do so, inspired by *spatialEval* (Cho et al., 2023b), we detect the spatial relation in images. However, we modify their approach to consider the given FoR when evaluating spatial relations. In particular, we convert all relations based on their FoR to be expressed from camera view and then

pass it to *spatialEval* evaluation since *spatialEval* assumes the camera perspective. We compare the bounding box and the depth map of two objects (i.e. relatum and locatum) mentioned in the spatial expression to determine the accuracy of the generated image. When evaluating the generated image from a context with FoR ambiguity, we consider it correct if it fits one of the valid FoRs for the given situation. See Figure 3 where context with FoR ambiguity produces two correct images in different FoR interpretations. We report the evaluation score in terms of VISOR$_{cond}$ and VISOR$_{uncond}$ (Gokhale et al., 2023). VISOR score is a metric designed to compare the spatial understanding abilities of T2I models. The VISOR$_{cond}$ evaluates the spatial relations and only includes the cases with both objects mentioned in the spatial expression correctly appearing in the generated image. In other words, it ignores cases with object errors and focuses on how well the model interprets spatial relations, which is the target of our work. While the VISOR$_{uncond}$ evaluates the model's overall performance, including object creation errors.

## 5.2 EXPERIMENTAL SETTING

**FoR Identification.** We selected five different LLMs including Llama3-8B, LLama3-70B (Llama, 2024), Gemma2-9B (Gemma, 2024), Qwen2-72B (Qwen Team, 2024), GPT-3.5-turbo (Brown et al., 2020), and GPT-4o (OpenAI, 2024) as the backbones for prompt engineering. The version of GPT-3.5-turbo is "gpt-3.5-turbo-0125," and GPT-4o is "gpt-4o-2024-05-13". We set the temperature of all models to be 0 to make the experiments reproducible. For each model, we apply several in-context learning (ICL) approaches including, *zero-shot*, *few-shot*, *CoT*, and our technique of Spatial-Guided Prompting (SG) as described in Section 4.1. For *few-shot*, *CoT*, and *SG*, we provide four examples to the models. The procedures for creating examples for each ICL are described in Section 4.1. The data splits used in these experiments are A-split and C-spilt.

**T2I.** We select Stable Diffusion 1.5 (SD-1.5) and Stable Diffusion 2.1 (SD-2.1) (Rombach et al., 2021) for stable diffusion models. For the backbone of layout-to-image, we choose GLIGEN (Li et al., 2023). We utilize LLama3-8B and LLama3-70B to handle the transition from spatial description to the textual bounding box information. The bounding box format is described in Section 4.2. To generate FoR information, we use the same selection of LLMs for the Spatial-Guided Layout Diffusion (SG Layout Diffusion), explained in Section 4.2. We generated four images per spatial expression to evaluate performance and calculated the VISOR score, following the original paper in Gokhale et al. (2023). The number of inference steps for all text-to-image models was set to 50. The data splits used in these experiments are I-A-split and I-C-split. For the evaluation, we select grounding DINO (Liu et al., 2024) and DPT (Ranftl et al., 2021), following VPEval Cho et al. (2023b), to detect objects and depth map, respectively. We conduct all experiments and evaluations on GPU A6000, taking roughly 300 GPU hours.

## 5.3 RESULTS

### 5.3.1 FOR INHERENTLY BIAS IN LLMS

**C-spilt.** The *zero-shot* setting reflects the LLMs' inherent bias in identifying FoR. Table 1 presents the accuracy for each FoR class in C-split, where sentences explicitly include information about topology and perspectives. We found that some models strongly prefer specific FoR classes. Notably, Gemme2-9B achieves a near-perfect accuracy on external relative FoR but performs poorly on other classes, especially external intrinsic, indicating a notable bias towards external relative. In contrast, GPT4o and Qwen2-72B show exceptional performance in both intrinsic FoR classes. However, they perform poorly in the relative FoRs.

**A-spilt.** We examine the FoR bias in the A-split. Based on the results in Table 1, we plotted the top-3 models' results (Gemma2-9B, Llama3-70B, and GPT4o) for a more precise analysis in Figures 4. The plots show the frequencies of each FoR category. According to the plot, Gemma and GPT have strong biases toward external relative and external intrinsic, respectively. This bias helps Gemma2 perform well in the A-split since all spatial expressions can be interpreted as external relative. However, GPT4o's bias leads to errors when intrinsic FoRs aren't valid, as in the Box and Pen cases (see plots (c) and (d)). Llama3 exhibits different behavior, showing a bias based on the relatum's properties, specifically the relatum's affordance as a container. In cases where relatum cannot serve as containers, i.e., Cow and Pen cases, Llama3 favors external relative. Conversely, Llama3 tends to favor external intrinsic when the relatum has the potential to be a container.

| Model | A-split | C-Split | | | | |
|---|---|---|---|---|---|---|
| | | ER-C-Split | EI-C-Split | II-C-Split | IR-C-Split | Avg. |
| Gemma2-9B (0-shot) | **94.17** | **94.24** | 35.98 | 53.91 | 57.66 | 60.45 |
| Gemma2-9B (4-shot) | 59.58 | 55.89(↓ 38.34) | 72.61(↑ 36.63) | 74.22(↑ 20.31) | 54.44(↓ 3.23) | 64.29(↑ 3.84) |
| Gemma2-9B (CoT) | 60.49 | 60.49(↓ 33.74) | 60.54(↑ 24.57) | 87.50(↑ 33.59) | 54.03(↓ 3.63) | 65.64(↑ 5.20) |
| Gemma2-9B (SG)(Our) | 72.67 | 65.87(↓ 28.37) | 65.54(↑ 29.57) | 53.12(↓ 0.78) | **95.97**(↑ 38.31) | 70.13(↑ 9.68) |
| llama3-8B (0-shot) | 59.58 | 60.36 | 83.80 | 56.25 | 62.50 | 65.73 |
| llama3-8B (4-shot) | 59.58 | 58.68(↓ 1.68) | 61.74(↓ 22.07) | 81.25(↑ 25.00) | 51.61(↓ 10.89) | 63.32(↓ 2.41) |
| llama3-8B (CoT) | 66.19 | 66.19(↑ 5.83) | 56.63(↓ 27.17) | 99.22(↑ 42.97) | 51.21(↓ 11.29) | 68.31(↑ 2.58) |
| llama3-8B (SG) (Our) | 72.73 | 69.88(↑ 9.52) | 49.24(↓ 34.57) | 100.00(↑ 43.75) | 49.19(↓ 13.31) | 67.08(↑ 1.35) |
| llama3-70B (0-shot) | 77.33 | 35.04 | 32.39 | 57.81 | 53.23 | 44.62 |
| llama3-70B (4-shot) | 59.78 | 59.78(↑ 24.74) | 66.52(↑ 34.13) | 77.34(↑ 19.53) | 51.61(↓ 1.61) | 63.81(↑ 19.20) |
| llama3-70B (CoT) | 66.00 | 68.01(↑ 32.97) | 65.65(↑ 33.26) | 91.41(↑ 33.59) | 58.47(↑ 5.24) | 70.88(↑ 26.27) |
| llama3-70B (SG) (Our) | 74.94 | 78.17(↑ 43.13) | 70.87(↑ 38.48) | 100.00(↑ 42.19) | 84.27(↑ 31.05) | 83.33(↑ 38.71) |
| Qwen2-72B (0-shot) | 60.21 | 60.21 | 93.70 | 85.16 | 45.16 | 71.06 |
| Qwen2-72B (4-shot) | 90.83 | 89.92(↑ 29.71) | 59.02(↓ 34.67) | 94.53(↑ 9.38) | 76.21(↑ 31.05) | 79.92(↑ 8.87) |
| Qwen2-72B (CoT) | 84.16 | 84.69(↑ 24.48) | 78.26(↓ 15.43) | 92.19(↑ 7.03) | 85.89(↑ 40.73) | 85.26(↑ 14.20) |
| Qwen2-72B (SG) | 93.84 | 92.93(↑ 32.72) | 97.39(↑ 3.70) | 96.09(↑ 10.94) | 85.08(↑ 39.92) | **92.87**(↑ 21.82) |
| GPT3.5 (0-shot) | 60.88 | 60.62 | 62.50 | 74.22 | 50.81 | 62.04 |
| GPT3.5 (4-shot) | 59.58 | 39.64(↓ 20.98) | 99.89(↑ 37.39) | 100.00(↑ 25.78) | 51.21(↑ 0.40) | 72.68(↑ 10.65) |
| GPT3.5 (CoT) | 59.13 | 59.52(↓ 1.10) | 74.67(↑ 12.17) | 100.00(↑ 25.78) | 48.39(↓ 2.42) | 70.65(↑ 8.61) |
| GPT3.5 (SG) (Our) | 77.59 | 69.62(↑ 9.00) | 97.93(↑ 35.43) | 100.00(↑ 25.78) | 60.48(↑ 9.68) | 82.01(↑ 19.97) |
| GPT4o (0-shot) | 59.90 | 60.43 | 99.35 | 100.00 | 51.61 | 77.85 |
| GPT4o (4-shot) | 59.78 | 59.91(↓ 0.52) | **100.00**(↑ 0.65) | 100.00 | 69.35(↑ 17.74) | 82.32(↑ 4.47) |
| GPT4o (CoT) | 64.31 | 63.99(↑ 3.56) | 99.89(↑ 0.54) | 100.00 | 62.10(↑ 10.48) | 81.49(↑ 3.65) |
| GPT4o (SG) (Our) | 69.88 | 70.08(↑ 9.65) | 99.67(↑ 0.33) | **100.00** | 73.39(↑ 21.77) | 85.78(↑ 7.94) |

Table 1: Accuracy results report from FoR Identification with LLMs. The correct prediction is one of the valid FoR classes for the given spatial expression. All FoR classes are external relative (ER), external intrinsic (EI), internal intrinsic (II), and internal relative (IR).

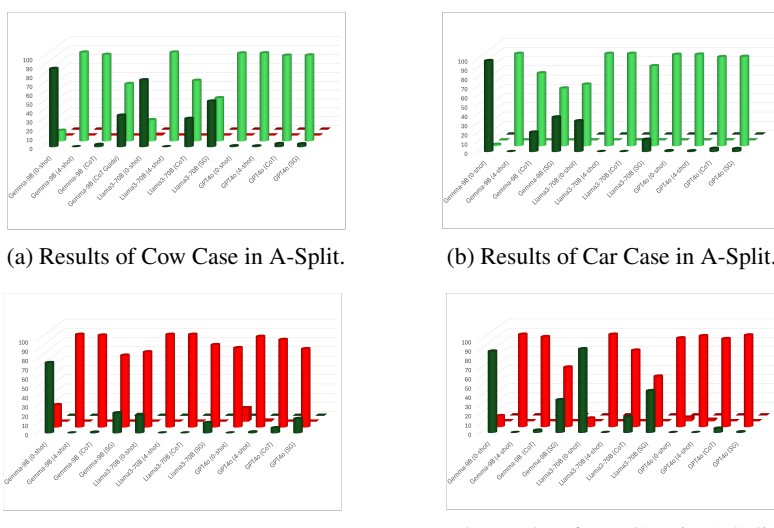

(a) Results of Cow Case in A-Split.    (b) Results of Car Case in A-Split.

(c) Results of Box Case in A-Split.    (d) Results of Pen Case in A-Split.

Figure 4: Red shows the wrong FoR identifications, and green shows the correct ones. The dark color is for relative FoRs, while the light color is for intrinsic FoRs. The round shape is for the external FoRs, while the square is for internal FoRs. The depth of the plots shows the four FoRs, i.e., *external relative, external intrinsic, internal intrinsic, and internal relative*, **from front to back**.

### 5.3.2 BEHAVIOR WITH ICL VARIATIONS

**C-spilt.** We evaluate the models' behavior under various in-context learning (ICL) methods. As observed in Table 1, the *few-shot* method improves the performance of the *zero-shot* method across multiple LLMs by reducing their original bias toward specific classes. Reducing the bias, however, lowers the performance in some cases, such as the performance of Gemma 2 in ER class. One noteworthy observation is that while the *CoT* prompting generally improves performance in larger LLMs, it is counterproductive in smaller models for some FoR classes. This suggests that the smaller models have difficulty inferring FoR from the longer context. This negative effect also appears in SG prompting, which uses longer explanations. Despite performance degradation in particular classes of small models, SG prompting performs exceptionally well across various models and achieves outstanding performance with Qwen2-72B. We further investigate the performance of CoT and SG prompting. As shown in Table 2, CoT exhibits a substantial difference in performance

| Model | inherently clear | | require template | |
|---|---|---|---|---|
| | CoT | SG | CoT | SG |
| Llama3-70B | 19.84 | 44.64 (↑ 24.80) | 76.72 | 87.39 (↑ 10.67) |
| Qwen2-72B | 58.20 | 84.22 (↑ 26.02) | 88.36 | 93.86 (↑ 10.67) |
| GPT-4o | 12.50 | 29.17 (↑ 16.67) | 87.73 | 90.74 (↑ 3.01) |

Table 2: The comparison between CoT and SG prompting in C-split separated by inherently clear / required template to be clear.

| | VISOR(%) | | | | | | | |
|---|---|---|---|---|---|---|---|---|
| | I-A-Split | | | | | I-C-Split | | |
| Model | cond (I) | | | cond (R) | cond (avg) | cond (I) | cond (R) | cond (avg) |
| | EI FoR | ER FoR | all | | | | | |
| SD-1.5 | 51.11 | 21.61 | 72.72 | 48.95 | 68.72 | 53.92 | 53.77 | 53.83 |
| SD-2.1 | 57.97 | 21.49 | 79.46 | 54.10 | 75.39 | **60.06** | 59.64 | 59.83 |
| Llama3-8B + GLIGEN | 53.67 | 25.78 | 79.45 | 66.08 | 77.38 | 57.51 | 65.98 | 62.12 |
| Llama3-70B + GLIGEN | 54.49 | 29.45 | 83.94 | 68.68 | 81.43 | 56.47 | 69.53 | 63.49 |
| Llama3-8B + SG + GLIGEN (Our) | 57.46 | 27.96 | 85.42 | **71.14** | 83.17 | 58.84 | **70.36** | **65.15** |
| Llama3-70B + SG + GLIGEN (Our) | 56.54 | 30.59 | **87.13** | 66.56 | **83.75** | 56.77 | 70.04 | 64.06 |

Table 3: VISOR$_{cond}$ score on the I-A and I-C splits where $I$ refer to the Cow case and Car case where relatum has intrinsic directions, and $R$ refer to the Box case and Pen case where relatum lacks intrinsic directions, $avg$ is mirco-average of $I$ and $R$. cond are explained in Section 5.1. EI and ER FoR represent the generated image considered corrected by EI or ER FoR

between contexts with inherently clear FoR and contexts requiring the template to clarify FoR ambiguity. This implies that CoT heavily relies on the specific template to identify FoR classes. In contrast, SG prompting demonstrates a smaller gap between these two scenarios and significantly enhances performance over CoT in inherently clear FoR contexts. Therefore, guiding the model to provide characteristics regarding topological, distance, and directional types of relations improves FoR comprehension. We provide failure examples of these two prompting methods in Appendix G.

**A-spilt.** We use the same Figure 4 to observe the behavior when applying ICL. The A-split shows minimal improvement with ICL variations, though some notable changes are observed. With *few-shot*, all models show a strong bias toward external intrinsic FoR, even when the relatum lacks intrinsic directions, i.e., Box and Pen cases. This bias appears even in Gemma2-9B, which usually behaves differently. This suggests that the models pick up biases from the examples despite efforts to avoid such patterns. However, *CoT* reduces some bias, leading LLMs to revisit relative, which is generally valid across scenarios. In Gemma2, the model predicts relative FoR where the relatum has intrinsic directions, i.e., Cow and Car cases. Llama3 behaves similarly in cases where the relatum cannot act as a container, i.e., Cow and Pen cases. GPT4o, however, does not depend on the relatum's properties and shows slight improvements across all cases. Unlike *CoT*, our SG prompting is effective in all scenarios. It significantly reduces biases while following a similar pattern to *CoT*. Specifically, SG prompting increases external relative predictions for Car and Cow in Gemma2-9B, and for Cow and Pen in Llama3-70B. Nevertheless, GPT4o shows only a slight bias reduction. However, Our proposed method improves the overall performance of most models, as shown in Table 1. The Llama3-70B behaviors are also seen in LLama3-8B and GPT3.5. The plots for these LLMs are in Appendix E due to lack of space.

### 5.3.3 FoR impact on Image Generation

We evaluate SG layout diffusion to assess the impact of using FoR on image generation. We focus on VISOR$_{cond}$ as it better reflects the model's spatial understanding than the overall performance measured by VISOR$_{uncond}$. Due to space limitations, VISOR$_{uncond}$ results are reported in Appendix D. Table 3 shows that adding FoR information (Llama3 + SG + GLIGEN) improves performance across all splits compared to the baseline models (Llama3 + GLIGEN). The most significant gains occur when the relatum lacks intrinsic direction, making external relative FoR the only valid option. However, the results show a significant bias towards the relative FoR of our model. This bias becomes more evident when comparing SD-2.1 with the baseline of our model (Llama3 + GLIGEN). This illustrates that the GLIGEN only significantly improves spatial comprehension on relative FoR. In contrast, SD-2.1 surpasses all GLIGEN-based models, including ours, when FoR is intrinsic, as seen in the cond($I$) of the I-C split in Table 3. This limitation likely arises from the reliance on

bounding boxes for generating spatial configurations, which makes it challenging to handle intrinsic FoR due to the lack of object properties and orientation. This challenge is further highlighted in the separate corrected interpretations for I-A split. From these results , GLIGEN only shows higher correct interpretation in external relative compared to SD-2.1. This confirms again that the main improvement in layout diffusion is in the relative FoR, which utilizes the camera perspective as co-ordinates for spatial relations. Regardless of GLIGEN's bias, incorporating FoR information from SG-prompting still improves all FoR classes. We provide further analysis of the improvement when employing SG in the layout generation in the Appendix F. Our experimental observations also show that Llama's bias when generating layouts aligns with the identified FoR, which prefers external intrinsic in A-Split and external relative in C-Split.

## 6 RELATED WORKS

Understanding situated spatial expressions requires knowledge of the frame of reference (FoR), which defines the coordinate system used to describe objects' positions. A detailed study of the FoR on multiple natural languages was conducted in (Levinson, 2003), which categorizes the FoR into three basic categories: intrinsic, relative, and absolute. Inspired by this basic framework, Tenbrink 2011 proposed a more comprehensive framework for specifying the FoR, used as the primary reference of our study. Their frameworks extended the basics with other spatial relation concepts, such as topology and temporal. Cognitive studies have increasingly focused on how humans perceive spatial FoR. Many findings in these studies suggest that humans favor specific FoR classes (Edmonds-Wathen, 2012; Vukovic & Williams, 2015; Shusterman & Li, 2016; Ruotolo et al., 2016) For instance, Ruotolo et al. 2016 investigated how the FoR affects the human's ability to memorize and describe the scene within a limited time. They found that participants were better at describing and answering questions when the spatial relations were based on participants' position, as opposed to using other objects as reference points. This highlights a gap between the relative and intrinsic FoR.

Several benchmarks have been developed across various domains to evaluate the spatial understanding of computation models. In the text-based domain, recent benchmarks focus on navigating with spatial instructions (Yamada et al., 2024) or question-answering tasks (Shi et al., 2022; Mirzaee & Kordjamshidi, 2022; Rizvi et al., 2024). These benchmarks are developed to assess the spatial reasoning capability without paying attention to FoR. Existing research often lacks explicit consideration of FoR, and the benchmarks do not include FoR annotations. Consequently, evaluating FoR understanding remains a research gap in spatial reasoning-related work. Similarly, text-to-image (T2I) benchmarks (Gokhale et al., 2023; Huang et al., 2023; Cho et al., 2023a;b) face the same issue. They usually focus on correctly placing two objects based on spatial relation from the camera perspective and relative FoR. Nevertheless, few works in vision-text domains are starting to recognize the importance of a FoR (Chen et al., 2024; Liu et al., 2023). One notable study is provided by Liu et al. 2023. They provide a case study on the FoR and results showing that making the model capable of understanding the FoR affects downstream performance on visual question answering. However, their study is limited in terms of FoR categories. In our work, we extend the coverage of benchmarks into more diverse frames of reference for the FoR recognition tasks. Moreover, we are the first to study the impact of FoR identification on text-to-image generation as a downstream task.

## 7 CONCLUSION

Given the significance of spatial reasoning in AI applications and the importance of understanding spatial frame of reference (FoR), we introduce **F**rame **o**f **R**eference **E**valuation in **S**patial Reasoning **T**asks (FoREST) benchmark to assess FoR comprehension in text-based spatial expressions and its impact on grounding in visual modality by diffusion models. Our benchmark results reveal notable differences in FoR identification in various LLMs. Moreover, the bias in FoR interpretations impacts the LLMs' ability to generate layouts for text-to-image generation. To improve FoR comprehension, we propose Spatial-Guided prompting, which guides the model in considering the type of spatial relations: topology, distance, and direction, resulting in more accurate FoR identification. This approach reduces the FoR biases in LLMs and improves the overall performance of the FoR identification task. Eventually, it enhances text-to-image generation performance by providing more accurate spatial configurations.

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

| Case | A-Split | I-A-Split | FoR class | C-Spilt | I-C-Spilt |
|------|---------|-----------|-----------|---------|-----------|
| Cow Case | 792 | 3168 | External Relative | 1528 | 4288 |
| Box Case | 120 | 120 | External Intrinsic | 920 | 3680 |
| Car Case | 128 | 512 | Internal Intrinsic | 128 | 0 |
| Pen Case | 488 | 488 | Internal Relative | 248 | 0 |
| Total | 1528 | 4288 | Total | 2824 | 7968 |

Table 4: Dataset Statistic of FoREST dataset.

| Category | Object | Intrinsic Direction | Container |
|----------|--------|---------------------|-----------|
| small object without intrinsic directions | umbrella, bag, suitcase, fire hydrant | ✗ | ✗ |
| bog object with intrinsic directions | bench, chair | ✓ | ✗ |
| big object without intrinsic direction | water tank | ✗ | ✗ |
| container | box, container | ✗ | ✓ |
| small animal | chicken, dog, cat | ✓ | ✗ |
| big animal | deer, horse, cow, sheep | ✓ | ✗ |
| small vehicle | bicycle | ✓ | ✗ |
| big vehicle | bus, car | ✓ | ✓ |
| tree | tree | ✗ | ✗ |

Table 5: All selected objects with two properties: intrinsic direction, affordance of being container

Jianing Yang, Xuweiyi Chen, Shengyi Qian, Nikhil Madaan, Madhavan Iyengar, David F. Fouhey, and Joyce Chai. Llm-grounder: Open-vocabulary 3d visual grounding with large language model as an agent. In *2024 IEEE International Conference on Robotics and Automation (ICRA)*, pp. 7694–7701, 2024. doi: 10.1109/ICRA57147.2024.10610443.

Yue Zhang and Parisa Kordjamshidi. Lovis: Learning orientation and visual signals for vision and language navigation, 2022.

## A DATASET STATISTICS

The FoREST dataset statistic is provided in the Table 4.

## B DETAILS CREATION OF FoREST DATASET

We define the nine categories of objects selected in our dataset as indicated below in Table 5. We select sets of locatum and relatum based on the properties of each class to cover four cases of frame of reference defined in Section 3.1. Notice that we also consider the appropriateness of the container; for example, the car should not contain the bus.

Based on the selected locatum and relatum. To create an A-split spatial expression, we substitute the actual locatum and relatum objects in the Spatial Relation template. After obtaining the A-split contexts, we create their counterparts using the perspective/topology clauses to make the counterparts in C-spilt. Then, we obtain the I-A and I-C split by applying the directional template to the first occurrence of relatum when it has intrinsic directions. The directional templates are "that is facing towards," "that is facing backward," "that is facing to the left," and "that is facing to the right." All the templates are in the Table 6. We then construct the scene configuration from each modified spatial expression and send it to the simulator developed using Unity3D. Eventually, the simulator produces four visualization images for each scene configuration.

### B.1 SIMULATION DETAILS

The simulation starts with randomly placing the relatum into the scene with the orientation based on the given scene configuration. We randomly select the orientation by given scene configuration, [-40, 40] for front, [40, 140] for left, [140, 220] for back, and [220, 320] for right. Then, we create the locatum from the relatum position and move it in the spatial relation provided. If the frame of reference is relative, we move the locatum based on the camera's orientation. Otherwise, we

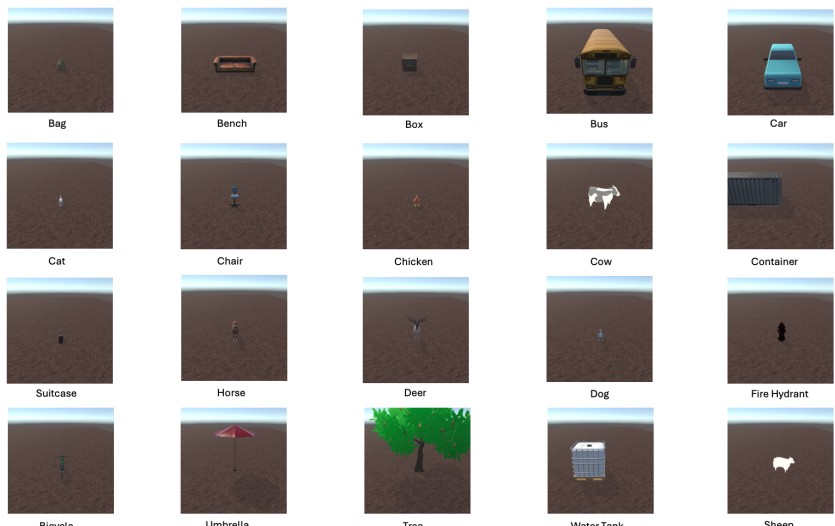

Figure 5: All 3d models used to generate visualizations for FoREST.

| | |
|---|---|
| Spatial Relation Templates | {locatum} is in front of {relatum}
{locatum} is on the left of {relatum}
{locatum} is to the left of {relatum}
{locatum} is behind of {relatum}
{locatum} is back of {relatum}
{locatum} is on the right of {relatum}
{locatum} is to the right of {relatum} |
| Topology Templates | within {relatum}
and inside {relatum}
and outside of {relatum} |
| Perspective Templates | from {relatum}'s view
relative to {relatum}
from {relatum}'s perspective
from my perspective
from my point of view
relative to observer |
| Directional Templates | that is facing toward
that is facing backward
that is facing to the left
that is facing to the right |

Table 6: All templates used to create FoREST dataset.

move it from the relatum's orientation. Then, we check the camera's visibility of both objects. If one of them is not visible, we repeat the process of generating the relatum until the correct placement is achieved. After getting the proper placement, we randomly choose the background from 6 backgrounds. Eventually, we repeat the procedures four times for one configuration.

## B.2 OBJECT MODELS AND BACKGROUND

For the object models and background, we find it from the unity assert store[2]. All of them are free and available for download. All of the 3D models used are shown in Figure 5.

---

[2]https://assetstore.unity.com

## B.3 TEXTUAL TEMPLATES

All the templates used to create FoREST are given in Table 6.

## C IN-CONTEXT LEARNING

We provide the prompting for each in-context learning. The prompting for *zero-shot* and *few-shot* is provided in Listing 1. The instruction answer for these two in-context learning is "Answer only the category without any explanation. The answer should be in the form of {Answer: Category.}"

For the Chain of Thought (CoT), we only modified the instruction answer to "Answer only the category with an explanation. The answer should be in the form of {Explanation: Explanation Answer: Category.}" Similarly to CoT, we only modified the instruction answer to "Answer only the category with an explanation regarding topological, distance, and direction aspects. The answer should be in the form of {Explanation: Explanation Answer: Category.}", respectively. The example responses are provided in Listing 2 for Spatial Guided prompting.

```
1  # Instruction to find frame of reference class of given context
2  """
3  Instruction:
4  You specialize in language and spatial relations, specifically in the
       reference frame of context. Identify the following context into the
       frame of reference categories (external intrinsic, internal intrinsic
       , external relative, internal relative) based on the information.
5
6  "External intrinsic is the context that uses spatial relation to describe
       the relative position of the object by referring to the reference
       object's direction, and both objects do not contain one another."
7
8  "Internal intrinsic is the context that uses spatial relation to describe
       the relative position of the object by referring to the reference
       object's direction and one object is inside another one"
9
10 "External relative is the context that uses spatial relation to describe
       the relative position of the object by referring to the observer's
       direction and both objects are in the same level, not contain one
       another."
11
12 "Internal relative is the context that uses spatial relation to describe
       the relative position of the object by referring to the observer's
       direction and one object is inside another one."
13
14
15 {Instruction answer}
16 # Normal Instruction answer: Answer without an explanation. The answer
       should be in the form of \{Answer: Category.\}
17 # COT Instruction answer: Answer only the category with an explanation.
       The answer should be in the form of \{Explanation: Explanation Answer
       : Category.\}
18 # SG Instruction answer: Answer only the category with an explanation
       regarding topological, distance, and direction aspects. The answer
       should be in the form of \{Explanation: Explanation Answer: Category
       .\}
19
20 Context: {spatial exprssion}
21
22 """
23
24 # Instruction for generate bounding box
25 """
26 Your task is to generate the bounding boxes of objects mentioned in the
       caption.
```

```
27   The image is size 512x512. The bounding box should be in the format of (x
        , y, width, height). Please considering the frame of reference of
        caption and direction of reference object if possible. If needed, you
         can make the reasonable guess.
28   """
```

Listing 1: Prompting to find the frame of reference class of given context and generate bounding box layout

```
1    ## This is example response used in few-shot, CoT, and SG
2    # Example 1: The bird is outside and in front of the car relative to the
        car.
3    # Few-shot response: Answer: external intrinsic
4    # CoT response: Explanation: The car, which is the relative (relatum)
        object, has the front/back direction, and the position of the bird
        refers to the front direction of the car based on the context. Answer
        : external intrinsic.
5    # SG response: Explanation: Topological: The bird is outside of the car.
        Distance: the bird is outside of the position of the car and should
        have some distance between the bird and the car. Direction: The car,
        the relative object, has the front/back direction, and the bird is
        actually in front of the car. Answer: external intrinsic
6
7
8    # Example 2: The bird is inside of the car and to the left of the car
        from the car's perspective.
9    # Few-shot response: Answer: internal intrinsic
10   # CoT response: Explanation: Based on the context, the bird is inside the
         car, and the position of the bird refers to the left part of the car
        . Answer: internal intrinsic.
11   # SG response: Explanation: Topological: bird is inside of the car.
        Distance: From the context, the bird should be in the same location
        as the car. Direction: The car, which is the relative object, has the
         front/back direction, and the bird is on the left side of the car.
        Answer: internal intrinsic.
12
13
14   # Example 3: The box is inside and at the back of the room.
15   # Few-shot response: Answer: internal relative
16   # CoT response: Explanation: The box is inside the room, but the room
        doesn't have the intrinsic direction. Therefore, the back relation is
         based on the observer's perspective of the room. Answer: internal
        relative.
17   # SG response: Explanation: Topological: The box is inside the room.
        Distance: box should be in the same location with the room from the
        context. Direction: The room doesn't have the direction and context
        reference from the observer's perspective, for the box is in the back
         of the room. Answer: internal relative.},
18
19
20   # Example 4: A phone is on the left of a tablet from my perspective.
21   # Few-shot response: Answer: external relative
22   # CoT response: Explanation: Based on the context, the tablet does not
        contain the phone. The tablet has the direction; however, the left of
         the tablet refers to my perspective rather than the tablet. Answer:
        external relative.
23   # SG response: Explanation: Topological: the phone is not inside of the
        tablet. Distance: The phone may have some distance from the tablet
        according to the context, but they should be near each other in the
        scene. Direction: even if the tablet has direction, the context left
        relation refers to the observer's perspective that a phone is on the
        left side of the tablet location. Answer: external relative
```

Listing 2: Spatial expression examples with the response for few-shots, Chain-of-Thought (CoT), and Spatial Guide (SG) prompting

| | VISOR(%) | | | | | |
|---|---|---|---|---|---|---|
| Model | uncond (I) | uncond (R) | uncond (avg) | uncond (I) | uncond (R) | uncond (avg) |
| | I-A-Split | | | I-C-Split | | |
| SD-1.5 | 45.43 | 33.22 | 43.51 | 35.06 | 35.68 | 35.40 |
| SD-2.1 | **62.87** | 43.90 | **59.89** | **45.98** | 46.59 | **46.31** |
| Llama3-8B + GLIGEN | 46.74 | 38.16 | 45.39 | 33.98 | 39.36 | 36.89 |
| Llama3-70B + GLIGEN | 54.33 | 46.89 | 53.17 | 38.04 | 46.04 | 42.37 |
| Llama3-8B + SG + GLIGEN (Our) | 51.83 | 43.24 | 50.48 | 36.28 | 44.43 | 40.70 |
| Llama3-70B + SG + GLIGEN (Our) | 58.92 | **47.44** | 57.12 | 38.23 | **48.62** | 43.86 |

Table 7: VISOR$_{uncond}$ score on the I-A-Split and I-C-Split where $I$ refer to the Cow Case and Car Case where relatum has intrinsic directions, and $R$ refer to the Box Case and Pen case where relatum lacks intrinsic directions, $avg$ is micro-average of $I$ and $R$. cond and uncond are explained in Section 5.1.

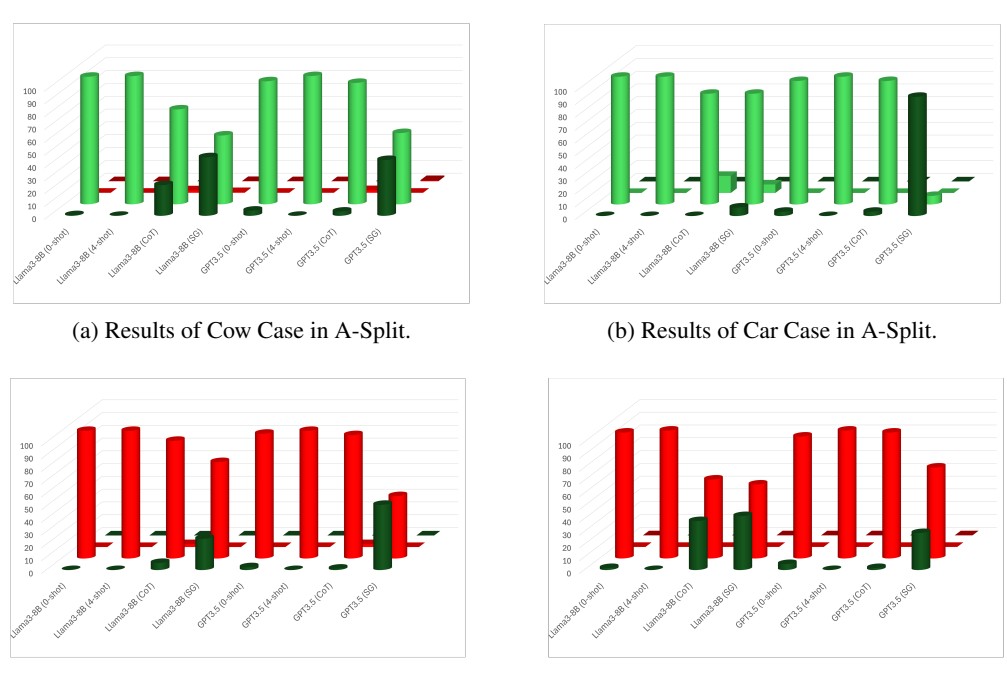

(a) Results of Cow Case in A-Split.  (b) Results of Car Case in A-Split.

(c) Results of Box Case in A-Split.  (d) Results of Pen Case in A-Split.

Figure 6: Red shows the wrong FoR identifications, and green shows the correct ones. The dark color is for relative FoRs, while the light color is for intrinsic FoRs. The round shape is for the external FoRs, while the square is for internal FoRs. The depth of the plots shows the four FoRs, i.e., external relative, external intrinsic, internal intrinsic, and internal relative, from front to back. This plot is the result of the rest of LLMs.

## D  VISOR SCORE

VISOR$_{uncond}$ provides the overall spatial relation score, including images with object generation errors. Since it is less focused on evaluating spatial interpretation than VISOR$_{cond}$, which assesses explicitly the text-to-image model's spatial reasoning, we report VISOR$_{uncond}$ results here in the Table 7 rather than in the main paper. The results are similar to the pattern observed in VISOR$_{uncond}$ that the based models(SD-1.5 and SD-2.1) perform better in the relative frame of reference, while the layout-to image models, i.e. GLIGEN, are better in the intrinsic frame of reference.

## E  A-SPLIT PLOT FOR FREQUENCY PREDICTIONS

Due to a lack of space and the same behavior as explained for LLama3-70B in Section 5.3.2, we provide the plot for Llama3-8B and GPT3.5 here in Table 6 instead of the main paper.

| Model | Layout | Layout$_{cond}$ |
|---|---|---|
| Llama3-8B | 85.26 | 88.84 |
| Llama3-8B + SG | 85.04 | 88.86 |
| Llama3-70B | 88.47 | 93.16 |
| Llama3-70B + SG | 91.95 | 95.45 |

Table 8: Layout accuracy where spatial relations are left or right relative to the camera. Layout is evaluated for all generated layouts in I-C split while Layout$_{cond}$ uses the same testing examples as VISOR$_{cond}$.

## F  ANALYZE THE IMPROVEMENTS IN SG-PROMPTING FOR T2I.

To further explain improvements of SG-prompting in T2I task, we assess the generated bounding boxes in the I-C split for left and right relations relative to the camera since these can be evaluated using only bounding boxes without depth information. As seen is Table 8, our SG prompting improved Llama3-70B's by $3.48\%$, while Llama3-8B saw a slight decrease of $0.22\%$. This evaluation was conducted on all generated layouts from the I-C split, which differs from the evaluation subset of images used for VISOR$_{cond}$ in Table 3. For a consistent evaluation, we report the layout$_{cond}$ score in the same table. Layout$_{cond}$ shows that Llama3-8B improves within the same evaluation subset with VISOR$_{cond}$. Overall, by incorporating FoR information through SG layout diffusion, Llama3 generates better spatial configurations, enhancing image generation performance.

## G  QUALITATIVE EXAMPLES OF FAILURE CASES OF COT AND SG

### G.1  COT INCORRECT BUT RESOLVED BY USING SG

- **Context** : A cat is inside and on the left of a car from the car's perspective

  **CoT** : Explanation: Cat is inside of the car, and the car has the intrinsic direction. However, the left of the car is referring to the car's perspective rather than the intrinsic direction of the car. Answer: internal relative.

  **SG** : Explanation: Topological: cat is inside of the car. Distance: cat should be in the same location with the car from the context. Direction: The car which is the relative object has the direction, and cat is on the left side of the car. Answer: internal intrinsic.

  **label** : internal intrinsic

- **Context** : A container is on the right of a dog from the dog's perspective

  **CoT** : Explanation: Based on the context, dog does not contain the container. The dog has the direction; however, the right of a dog referring to the dog's perspective rather than intrinsic direction of the dog. Answer: external relative.

  **SG** : Explanation: Topological: container is not inside of the dog. Distance: container may has some distance from dog according to the context. Direction: Dog has the direction and the context right relation is referred from dog's perspective that a container is on the right side of the dog's facing orientation. Answer: external intrinsic.

  **label** : external intrinsic

- **Context** : A water tank is on the left of a cow from the cow's perspective

  **CoT** : Explanation: Based on the context, cow does not contain the water tank. The cow has the direction; however, the left of a water tank referring to the cow's perspective rather than cow's intrinsic direction. Answer: external relative.

  **SG** : Explanation: Topological: water tank is not inside of the cow. Distance: water tank may has some distance from cow according to the context. Direction: The cow which is the relative object has the direction, and water tank is on the left side of the cow. Answer: external intrinsic.

  **label** : external intrinsic

- **Context** : A bicycle is in front of a container inside the container

  **CoT** : Explanation: The bicycle is contained within the container, fulfilling the first condition for internal frames of reference. However, since the bicycle is also described as

being 'in front' of another object (another container), this implies a spatial relationship that is dependent on the container's intrinsic direction. But due to thembiguity of having two containers mentioned, and assuming the second 'container' is a typo or misstatement, we interpret it as a single container scenario. As such, the spatial relation is intrinsic to the container itself, not relative to an external observer or another entity. Answer: internal intrinsic.

However, the context provided contains a potential error or ambiguity ("inside the container"), which could alter the interpretation if clarified. Assuming there was meant to be only one container, thebove categorization holds. If there were indeed two distinct containers, further clarification would be needed to accurately categorize the spatial relation.

**SG** : Explanation: Topological: bicycle is inside of the container. Distance: bicycle should be in the same location with the container from the context. Direction: Container doesn't have the direction and context refere from observer perspective for bicycle is in the front of the container. Answer: internal relative. However, there seems to be a contradiction in the statement as it says the bicycle is both inside and in front of the container. For the purpose of categorization, we will assume that the intended meaning was that the bicycle is inside the container and positioned towards the front end of it.

**label** : internal relative

- **Context** : A tree is on the left of a box inside the box

  **CoT** : Explanation: Based on the context, box does not contain the tree. However, it seems there might be a mistake in the sentence as it mentions 'inside the box' twice. Assuming the intention was to describe the tree's position relative to the box without specifying a container for the tree, we can interpret this as the tree being positioned relative to the box without being contained within it. Since the box does not have an intrinsic direction, the reference to 'left' is relative to an external observer's perspective. Answer: external relative.

  **SG** : This context seems to have a logical inconsistency as a tree cannot be inside a box. However, assuming this was intended to describe two separate spatial relations, we can interpret it as follows:

  Explanation: Topological: tree is not inside of the box. Distance: tree may have some distance from the box according to the context. Direction: the box does not have a facing orientation, so the tree being on the left is referred from the observer's perspective. Answer: external relative for the relation between the tree and the box.

  However, the second part of the sentence "inside the box" does not provide enough information to determine a spatial relation between entities. It appears to be incomplete or incorrectly phrased. If you meant to say "A tree is on the left of a box inside a room," then the relation between the tree and the box would still be external relative, while the relation of the box being inside the room would be internal relative.

  **label** : internal relative

- **Context** : A deer is outside and on the left of a bus relative to the bus

  **CoT** : Explanation: The bus has the front/back direction. However, the left of the bus is not referring to the bus's intrinsic direction, but rather to an observer's perspective. Answer: external relative.

  **SG** : Explanation: Topological: deer is outside of the bus. Distance: deer may be little bit far from the left of the bus from the context. Direction: The bus which is the relative object has the direction and deer is actually on the left side of the bus. Answer: external intrinsic.

  **label** : external intrinsic

- **Context** : A bicycle is behind a fire hydrant

  **CoT** : Explanation: Based on the context, fire hydrant does not contain the bicycle. Fire hydrant does not have intrinsic direction, so the behind relation is based on the observer's perspective. However, since there is no explicit mention of an observer or reference point, we can consider it as an external intrinsic relation where the observer's perspective is implied. Answer: external intrinsic.

  **SG** : Explanation: Topological: bicycle is not inside of the fire hydrant. Distance: bicycle may have some distance from the fire hydrant according to the context. Direction: Fire hy-

drant does not have a facing orientation, so the spatial relation is not relative to its intrinsic direction. Instead, it is described from an external perspective. Answer: external relative.

**label** : external relative

- **Context** : A box is behind a chicken relative to the chicken

  **CoT** : Explanation: The chicken has no intrinsic direction, so the 'behind' relation is based on the perspective of the chicken itself, not its intrinsic orientation. Answer: external relative. Note that this could potentially be seen as external intrinsic if we assume that the chicken has a front-back orientation, but since chickens do not typically have a fixed or understood front-back orientation like humans or cars do, it is more likely to be considered external relative.

  **SG** : Explanation: Topological: box is not inside of the chicken. Distance: box may be little bit far from the chicken from the context. Direction: The chicken which is the relative object has the direction and box is actually behind the chicken. Answer: external intrinsic.

  **label** : external intrinsic

- **Context** : A car is in front of a deer from the deer's perspective

  **CoT** : Explanation: Based on the context, deer does not contain the car. The deer has the direction; however, the front of a deer referring to the deer's perspective rather than intrinsic direction of deer. Answer: external relative.

  **SG** : Explanation: Topological: car is not inside of the deer. Distance: car may has some distance from deer according to the context. Direction: The deer which is the relative object has the direction, and car is in front of the deer. Answer: external intrinsic.

  **label** : external intrinsic

- **Context** : An umbrella is on the left of a chicken from the chicken's perspective

  **CoT** : Explanation: Based on the context, chicken does not contain the umbrella. The chicken has the direction; however, the left of a chicken referring to the chicken's perspective rather than intrinsic direction of the chicken. Answer: external intrinsic. Note that this sentence is ambiguous because it can be interpreted as external relative if the left is referring to the observer's perspective. However, given the context provided, it is more likely to be interpreted as external intrinsic.

  **SG** : Explanation: Topological: umbrella is not inside of the chicken. Distance: umbrella may has some distance from chicken according to the context. Direction: The context left relation is referred from the chicken's perspective that an umbrella is on the left side of the chicken's facing orientation. Answer: external intrinsic.

  **label** : external intrinsic

### G.1.1 FAILURE CASE OF SG

- **Context** : A bus is on the right of a chair from the chair's perspective

  **SG** : Explanation: Topological: bus is not inside of the chair. Distance: bus may has some distance from chair according to the context. Direction: Even though the chair does not have a facing orientation, the perspective is given as the chair's, making this a case of an object being described relative to another object's perspective. Answer: external intrinsic. However, usually chairs do not have a defined facing direction, so typically this would fall under external relative. But given the instruction to consider the chair's perspective, it can be seen as an exception to the rule, aligning more closely with external intrinsic in this specific context.

  **label** : external intrinsic

- **Context** : A tree is on the left of a box inside the box

  **SG** : This context seems to have a logical inconsistency as a tree cannot be inside a box. However, assuming this was intended to mean 'A tree is on the left of a box inside a room', the categorization would be:

  Explanation: Topological: tree is not inside of the box. Distance: tree may have some distance from the box according to the context. Direction: The context left relation is referred from observer's perspective that a tree is on the left side of the box location. Answer: external relative.

| Model | ER | | EI | | II | | IR | |
|---|---|---|---|---|---|---|---|---|
| | temp-0 | temp-1 | temp-0 | temp-1 | temp-0 | temp-1 | temp-0 | temp-1 |
| Cow Case | | | | | | | | |
| 0-shot | 75.38 | 87.12 | 23.86 | 12.50 | 0.76 | 0.13 | 0.00 | 0.25 |
| 4-shot | 0.00 | 15.66 | 100.00 | 84.34 | 0.00 | 0.00 | 0.00 | 0.00 |
| CoT | 31.82 | 49.87 | 68.18 | 49.87 | 0.00 | 0.13 | 0.00 | 0.13 |
| SG | 51.39 | 70.45 | 48.61 | 29.42 | 0.00 | 0.00 | 0.00 | 0.13 |
| Box Case | | | | | | | | |
| 0-shot | 22.50 | 41.67 | 77.50 | 58.33 | 0.00 | 0.13 | 0.00 | 0.25 |
| 4-shot | 0.00 | 0.00 | 100.00 | 100.00 | 0.00 | 0.00 | 0.00 | 0.00 |
| CoT | 0.00 | 5.83 | 100.00 | 94.17 | 0.00 | 0.00 | 0.00 | 0.00 |
| SG | 11.67 | 33.33 | 88.33 | 66.67 | 0.00 | 0.00 | 0.00 | 0.00 |
| Car Case | | | | | | | | |
| 0-shot | 55.20 | 68.24 | 49.01 | 31.15 | 0.79 | 0.61 | 0.00 | 0.00 |
| 4-shot | 0.60 | 5.94 | 99.40 | 94.06 | 0.00 | 0.00 | 0.00 | 0.00 |
| CoT | 19.64 | 38.52 | 80.16 | 61.27 | 0.20 | 0.20 | 0.00 | 0.00 |
| SG | 44.25 | 56.97 | 55.75 | 43.03 | 0.00 | 0.00 | 0.00 | 0.00 |
| Pen Case | | | | | | | | |
| 0-shot | 90.62 | 96.88 | 9.38 | 3.12 | 0.00 | 0.61 | 0.00 | 0.00 |
| 4-shot | 0.00 | 7.03 | 100.00 | 92.97 | 0.00 | 0.00 | 0.00 | 0.00 |
| CoT | 17.19 | 28.91 | 82.81 | 71.09 | 0.20 | 0.20 | 0.00 | 0.00 |
| SG | 48.31 | 57.81 | 54.69 | 42.19 | 0.00 | 0.00 | 0.00 | 0.00 |

Table 9: The results between two different temperatures of Llam3-70B on the A-spilt of FoREST. The number shows the percentage frequency of responses from the model.

> If the original context was correct, it would fall under none of these categories due to the physical impossibility of the scenario described.
>
> **label** : internal relative

- **Context** : A cow is in front of a bench relative to the bench

  **SG** : Explanation: Topological: cow is not inside of the bench. Distance: cow may has some distance from bench according to the context. Direction: Bench doesn't have the direction and context refere from observer perspective for cow is in the front of the bench. Answer: external relative. But if we consider that 'in front of' refers to the bench's intrinsic direction, then it would fall under external intrinsic. However, benches typically do not have a defined facing direction, so this is less common.

  **label** : external intrinsic

- **Context** : A dog is on the left of a box within the box

  **SG** : Explanation: Topological: dog is not inside of the box. Distance: dog may has some distance from box according to the context. Direction: Even if box has direction, the context left relation is referred from observer's perspective that a dog is on the left side of the box location. However, since the dog is not contained by the box, this does not fall under internal intrinsic or internal relative. Answer: external relative. Note that the phrase "within the box" seems redundant and does not change the frame of reference.

  **label** : internal relative

# H EXPERIMENT WITH DIFFERENT TEMPERATURES

To further investigate the impact of temperature on the biased interpretation of the model in the A-split of our dataset, we conducted additional experiments. As presented in Table 9, comparing distinct temperatures (0 and 1) revealed a shift in the distribution. The frequencies of the classes experienced a change of up to 10%. However, the magnitude of this change is relatively minor, and the relative preferences for most categories remained unchanged. Specifically, the model exhibited the highest frequency responses for the cow, car, and pen cases, even with higher frequencies in certain settings. Consequently, a high temperature does not substantially alter the diversity of LLMs' responses to this task, which is an intriguing finding.

