# OpenReview forum: "FoREST: Frame of Reference Evaluation in Spatial Reasoning Tasks"
_ICLR.cc/2025/Conference — Submitted to ICLR 2025_

### Official Review · Reviewer_bEBN · 2024-10-28

**Soundness:** 3
**Presentation:** 2
**Contribution:** 2
**Rating:** 3
**Confidence:** 4

**Summary:**

Research on spatial perception capabilities in large language models is a key direction for optimizing their generation abilities. This paper introduces a benchmark for understanding frames of reference (FoR) and evaluates different LLMs to test the spatial perception capabilities. Additionally, it uses diffusion models to conduct experimental visualizations that simulate this understanding. The benchmark provides guidance on designing prompts that enhance spatial reasoning, contributing to improved text-to-image accuracy.

**Strengths:**

This paper focuses on a core shortcoming in large language models' text comprehension-spatial understanding, presenting a benchmark for evaluating spatial perception abilities. It summarizes potential situations in spatial perception using existing evaluation metrics. Downstream tasks incorporate text-to-image generation experiments with diffusion models to visualize the difference in spatial understanding capabilities of different LLMs. The paper is well-structured, with straightforward explanations of the methods, and employs vivid examples to enhance understanding.

**Weaknesses:**

However, this paper lacks logical coherence in its descriptions of various cases. The selection of content for the four FoR classes seems random, raising the question of whether there could be a clearer division for categorizing different spatial reasoning tasks. It remains unclear if the four cases can comprehensively cover all possible spatial reasoning tasks, and more citations are needed to support your claim. The presentation of experimental results is also insufficient; for instance, the two images in Figure 3 fail to follow the principle of controlling variables, rendering the conclusions unconvincing. Furthermore, the experimental analysis is inadequate; while the comparison between LLaMA3-8B and LLaMA3-70B results is noteworthy, the spatial understanding of LLaMA declines as parameters increase, varying across C-Split cases. This raises the question of what insights researchers can draw from this benchmark to adjust datasets to maintain or even improve spatial understanding performance. Addressing this issue is essential to the benchmark's purpose.

**Questions:**

I would look forward to seeing more experimental results, particularly on how different large language models perform on the benchmark. For example, how do the latest models like Qwen2 or Molmo perform in different cases? Additionally, it’s intriguing that spatial understanding may decline as parameter count increases—what could be the underlying reasons? I also noticed that the performance decreases with the use of CoT and 4-shot settings, which is puzzling. What might be causing this effect?

---

> ### Author Response · Authors · 2024-11-23
>
> We appreciate your valuable feedback on our work. We want to address each weakness you mentioned and the question below.
>
> ## Randomness in the selection of four frame-of-reference (FoR) classes
>
> We respectfully disagree with the comment on the randomness of our selection.
> Our selection of four frames of reference classes is based on our deep study of the related work in linguistics, as we have cited a few in the paper [1, 2]. some of the reviewers also highlighted the soundness of our choice and the theoretical support. Similar terms are used across several cognitive studies.
> One example is in [3]. They used the terms egocentric and allocentric, which have the same meaning as relative and intrinsic frames of reference, respectively.
> The related work section provides more examples of using similar terms.
> For the AI community, we also encounter some literature that use similar concepts but may be in different terms.
> We provide two highly relevant references in the related work section.
> The first one [4] uses the same terms for the frame of reference, intrinsic and relative, while the other [5] uses object-centric terms to represent the same concept as an intrinsic frame of reference. Also, below, we highlighted the papers we have already cited in the paper to support our claims regarding our choice of FOR.
>
>
> [1] Stephen C. Levinson. Space in Language and Cognition: Explorations in Cognitive Diversity. Language Culture and Cognition. Cambridge University Press, 2003.
>
> [2] Thora Tenbrink. Reference frames of space and time in language. Journal of Pragmatics, 43(3):704–722, 2011. ISSN 0378-2166. doi: https://doi.org/10.1016/j.pragma.2010.06.020. URL https://www.sciencedirect.com/science/article/pii/S037821661000192X. The Language of Space and Time.
>
> [3] Francesco Ruotolo, Tina Iachini, Gennaro Ruggiero, Ineke J. M. van der Ham, and Albert Postma. Frames of reference and categorical/coordinate spatial relations in a “what was where” task. Experimental Brain Research, 234(9):2687–2696, Sep 2016. ISSN 1432-1106. doi: 10.1007/s00221-016-4672-y. URL https://doi.org/10.1007/s00221-016-4672-y.
>
> [4] Fangyu Liu, Guy Emerson, and Nigel Collier. Visual spatial reasoning, 2023. Transactions of the Association for Computational Linguistics.
>
> [5] Boyuan Chen, Zhuo Xu, Sean Kirmani, Brian Ichter, Danny Driess, Pete Florence, Dorsa Sadigh, Leonidas Guibas, and Fei Xia. Spatialvlm: Endowing vision-language models with spatial reasoning capabilities, 2024. Proceedings of the IEEE/CVF Conference on Computer Vision and Pattern Recognition (CVPR).
>
> ## Figure 3 issue
>
> We would appreciate it if the reviewer could rephrase their question and clarify which conclusion is not convincing. Our conclusion from this part of our research is that characterizing the frame of reference is helpful for T2I. However, when the FoR is inherently ambiguous, multiple valid images exist, and if we can characterize the FoR as much as possible, then we can get one of those valid images.
>
> Figure 3 only illustrates an ambiguity case in spatial expressions in the A-split and the possible correct images.
> In “A car is to the right of a cow,” the car can be positioned to the cow's actual right or the right of the cow’s location from the camera's perspective.
> We consider both options as valid interpretations in the A-split.
> Conversely, the counter-part context in C-split, such as “a car to the right of a cow from the camera’s perspective,” has only one correct interpretation, corresponding to image(b) in Figure 3.
>
> ## Experimental analysis of Llama results
> Thank you for the interesting question. In fact, we think our results are insightful for understanding the original bias of the LLMs at different sizes. Llama3-70 B's lower performance is observed in the zero setting, where we lack control over the model’s behavior.
> One explanation is that the larger models acquire stronger biases from their training examples and memorize the FoR patterns.
> However, the larger model generally exhibits superior instruction-following abilities and mitigates this issue when additional instructions are provided to elaborate on the answer in more sophisticated prompt-engineering settings.
> This is evident in the improvements observed when employing CoT and SG-prompting.

---

> ### Author Response · Authors · 2024-11-23
>
> ## Additional results
>
> Thank you for your comment, we conducted the additional experiments, reported here on Qwen2.
> Since Molmo is a multi-model model based on Qwen2, we anticipate its performance to be comparable.
> We observed that the 7B variance of Qwen2's performance is equivalent to that of Gemma2-9B.
> Conversely, the 72B variant of Qwen2 exhibits a distinct behavior compared to the other models utilized in our dataset.
> The model's default (zero-shot setting) interpretation aligns with the GPT family (prefer external intrinsic cases).
> However, when employing a few-shot setting or CoT, the model prefers external relative cases over external intrinsic cases, resulting in exceptionally high performance.
> This is because the model assumes that most objects do not have front and back on their own and that the spatial relation is created from an outside perspective, which is opposite to the GPT.
> Nevertheless, this assumption lowers the performance in the EI of C-split.
> Our SG prompting seems to resolve this issue, improving performance in this category and helping the model achieve SOTA results in C-split.
>
> | Model               | ER-C-Split        | EI-C-Split        | II-C-Split        | IR-C-Split        | Avg               | A-split          |
> |---------------------|-------------------|-------------------|-------------------|-------------------|-------------------|------------------|
> | Qwen2-7B (0-shot)   | 99.61             | 2.07              | 35.94             | 24.60             | 40.55             | 99.34            |
> | Qwen2-7B (4-shot)   | 34.36 ↓(65.25)    | 65.11 ↑(63.04)    | 89.84 ↑(53.91)    | 89.52 ↑(64.92)    | 69.71 ↑(29.16)    | 61.71            |
> | Qwen2-7B (CoT)      | 53.40 ↓(46.20)    | 78.59 ↑(76.52)    | 100.00 ↑(64.06)   | 49.60 ↑(25.00)    | 70.40 ↑(29.85)    | 61.38            |
> | Qwen2-7B (SG)       | 71.53 ↓(28.08)    | 79.46 ↑(77.39)    | 96.88 ↑(60.94)    | 59.27 ↑(34.68)    | 76.78 ↑(36.23)    | 73.30            |
> | Qwen2-72B (0-shot)  | 60.21             | 93.70             | 85.16             | 45.16             | 71.06             | 60.21            |
> | Qwen2-72B (4-shot)  | 89.92 ↑(29.71)    | 59.02 ↓(34.67)    | 94.53 ↑(9.38)     | 76.21 ↑(31.05)    | 79.92 ↑(8.87)     | 90.83            |
> | Qwen2-72B (CoT)     | 84.69 ↑(24.48)    | 78.26 ↓(15.43)    | 92.19 ↑(7.03)     | 85.89 ↑(40.73)    | 85.26 ↑(14.20)    | 84.16            |
> | Qwen2-72B (SG)      | 92.93 ↑(32.72)    | 97.39 ↑(3.70)     | 96.09 ↑(10.94)    | 85.08 ↑(39.92)    | 92.87 ↑(21.82)    | 93.84            |
>
> *Table: Results of Qwen2-7B and Qwen2-72B on our dataset. "↑(number)" indicates improvement over 0-shot by (number), and "↓(number)" indicates decrease compared to 0-shot by (number).*
>
> ## LLama3 results
>
> One potential explanation is that the larger models acquire biases from their training examples, which can lead to confusion in zero-shot experiments. In these experiments, we lack control over the model’s behavior except for the prompt, which remains constant across all settings.
> This issue can be mitigated when the model provides additional information in CoT and SG prompting, as illustrated in Table 1.
> Conversely, the smaller model experiences a decline in performance when additional explanation is necessary. One plausible explanation for this observation is that the model provided an erroneous interpretation, resulting in an inaccurate conclusion. This phenomenon may be attributed to the extended generated sequence, as the larger model does not encounter this issue.
>
>
> We hope that all responses address any confusion the reviewer raised. We hope they are convincing about our work and could increase our score. Again, we really appreciate your comment. We will let you know again when we upload the revised version.

---

> > ### Comment · Reviewer_bEBN · 2024-11-25
> >
> > Thanks to the authors for the detailed response. The explanation of four FoR classes setting is clearer and experiments about qwen is convincing to prove the necessity of SG.
> >     The question about figure 3 is why you use different generation models to show your result. "If FoR is inherently ambiguous, multiple valid images exist". How do you show this case? I only see a wrong image generated by SD-2.1. "If we can characterize the FoR as much as possible, then we can get one of those valid images". This image is from " Llam3-8B + GLIGEN"? not SD-2.1? Then how can I see the importance of FoR?
> >      The experimental results of Qwen-7b are consistent with my understanding of prompt engineering technology, and the improvement in 4-shot and CoT confirms this. I have some doubts about the validity of the experimental results of llama-8b, but it does not affect the proof of the innovation of this work. I did not find the prompt (only example) you used for CoT and SG in the supplementary materials, and I hope you can also supplement it.

---

> > > ### Author Response · Authors · 2024-11-25
> > >
> > > >why you use different generation models to show your result:
> > >
> > > SD-2.1 consistency generated the right image using the COW intrinsic right. We wanted to show the possible variety of the solutions, that is why we used the output of multiple models. We hope this makes sense now.
> > >
> > > >SD-2.1 correctness:
> > >
> > > The image generated by SD-2.1 is **actually correct**, if you think about it from the Cow perspective and the intrinsic right side of the COW. We are not sure what is the confusion point, can you please clarify further?
> > >
> > > >how can I see the importance of FoR?
> > >
> > > In Figure 3 we just showed two cases that are both correctly generated without our prompting effort with FoR information. We will add an example for the wrong cases generated by the same models and will demonstrate they how it was fixed and generated the valid image considering the FoR.
> > >
> > > > LLama-* results vs Qwen-* results
> > >
> > > To our understanding all the results are consistent when we look at it as follows,
> > >   - Zero-shot settings reflect the original bias of the models and depending on that bias even large models can have lower accuracy compared to small models. This depends on their more training that can increase a  certain bias for large models.  Holds for both LLama and Qwen.
> > >  - COT and 4-shot: Alway increase the performance of large models significantly, due to their ability to follow instructions with a larger context. The impact is not alway good for smaller models. Holds both for LLama and Qwen.
> > >  - SG model: sharply improves the large models and most of the time better than COT.
> > > Please let us know which part is inconsistent and why you think LLama-8b does not sound valid.
> > >
> > > > the actual COT and SG prompts:
> > >
> > > We have a full example in page 15 of appendix, and in that case the place holder of {instruction answer} contains the actual text that we sho isn 752-754 for COT and 754 and 778-779. We replace that content in the full example for the clarity in the appendix.

---

> > > > ### Comment · Reviewer_bEBN · 2024-11-27
> > > >
> > > > Thanks to the author for the feedback. Some of my concerns have been solved, but some still exist.
> > > >        1.I understand what you want to express in Figure 3, and I agree it is necessary to add error examples to show the importance of FoR.
> > > >        2. The result of Llama. I think the authors are avoiding my question. Even if CoT has a smaller impact on small models, it should not have a negative impact on the experimental results. Through the details given in the appendix, I judge that the quality of  the prompt for CoT is ill-considered. Using the rough CoT as a baseline and presenting the carefully designed SG as their own method will diminish the rigor of this paper in my evaluation. This leads me to decrease my score.

---

### Official Review · Reviewer_WWK9 · 2024-10-29

**Soundness:** 3
**Presentation:** 3
**Contribution:** 2
**Rating:** 3
**Confidence:** 4

**Summary:**

This paper introduces a new benchmark called FoREST to evaluate LLMs spatial ability in understanding "Frame of Reference". Additionally, it proposes Spatial-Guided prompting to improve the FoR classification task and layout generation in text-to-image generation.

**Strengths:**

1. Spatial ability of LLMs are an important research topic yet less explored. A scientific benchmark would contribute to this area.
2. This work conducts various experiments with a range of LLM models and provide in-depth analysis. It also verifies the proposed prompting method in text-to-image task, adding its value to real world applications.
3. The paper is well organized and written.

**Weaknesses:**

1. The dataset is pure synthetic and constructed by a limited number of textual templates. I have concerns about the FoR classification task given the template "<locatum> <spatial relation> <relatum> <perspective>". It seems hard to disentangle this task with linguistic and common-sense reasoning of LLMs. For example,  LLMs are able to determine whether the perspective is intrinsic or relative by analyzing perspective template, and analyze topology template to determine whether the locatum is external or internal. Both of them don't necessitate understanding the underlying spatial configuration under a specific perspective. On the contrary, the text-to-image task indeed requires the model to interprete the spatial configuration and transform the perspective to camera's.
2. Again, since the dataset is synthetic and constructed by textual templates, the inductive bias might be leveraged by SG-prompting.
3. Lack of full prompts of different settings, such as few-shot, CoT, SG-prompting, text-to-layout and SG to layout.

**Questions:**

1. Is SG-prompting zero-shot or few-shot? In T2I task, what are examples mentioned in line 290?
2. In table 4, it's abnormal to see GPT-3.5 outperforms GPT-4o in A-split. And why do GPT family models do exceptionally well in EI and II C-split, yet relatively bad at ER and IR C-split? It would be interesting to dive into these observations.

---

> ### Author Response · Authors · 2024-11-24
> **Response (1)**
>
> We appreciate your valuable feedback on our work. We want to address each weakness you mentioned and the question below.
>
>
> # Weaknesses
>
> ## Concern about using the template
>
> Thank you for raising the issue of the textual template. While we agree that it is a restricted template, we clarify the motivation and explain why it is still interesting.
>
> Given the language's ambiguity, we need to add more information, such as perspectives and topology, to create a split of the dataset with unambiguous FoR classes.
> This information in the template helps the language model to use this information for spatial inference/reasoning.
> The templates are provided to characterize the essential elements/concepts that should be used for spatial reasoning.
> Though this simplifies reasoning if the language model can capture the pattern, our experiments show that it can still fail to recognize the frame of reference (see Example 1,2,3 (CoT fails and resolved by SG) and  10, 11, 12, 15 (SG failure)).
>
> Moreover, some spatial expressions are inherently explicit and without ambiguity.
> We do not need to add the perspective and topology information following the template for those cases. In other words, a proportion of our data does not follow the template, and the model must also classify the FoR for those unambiguous cases.
>
> Our results show that SG Prompting improves the FoR classification significantly when there is no template; see the table below and the qualitative example below. Also, as you pointed out, if we can make the FoR-related concepts explicit, the models can accurately output the FoR, and the text-to-image generation models can use that information for accurate visualization.
>
> | Model      | CoT (no template) | SG (no template) | CoT (with template) | SG (with template) |
> |------------|-------------------|------------------|---------------------|--------------------|
> | Gemma-9B   | 2.58              | 35.51 (↑ 32.93)  | 72.65               | 73.80 (↑ 1.15)     |
> | Llama3-8B  | 22.22             | 36.90 (↑ 14.68)  | 73.64               | 71.07 (↓ 2.57)     |
> | Llama3-70B | 19.84             | 44.64 (↑ 24.80)  | 76.72               | 87.39 (↑ 10.67)    |
> | Qwen2      | 58.20             | 84.22 (↑ 26.02)  | 88.36               | 93.86 (↑ 5.50)     |
> | GPT-3.5    | 1.58              | 43.25 (↑ 41.67)  | 77.64               | 85.21 (↑ 7.57)     |
> | GPT-4o      | 12.50             | 29.17 (↑ 16.67)  | 87.73               | 90.74 (↑ 3.01)     |
>
> *Table: The table presents the results of our C-Split experiment on our dataset. The “↑” symbol indicates an improvement over the CoT baseline, while the “↓” symbol denotes a decrease compared to the CoT baseline. The table is divided into two sections: one for context with templates and another for context without templates. It is important to note that the context without templates is inherently clear and does not require additional information.*
>
> **We will include the table in the main paper and examples in the appendix for the qualitative analysis of our experiment**
>
> Failure examples are in the following response.

---

> ### Author Response · Authors · 2024-11-24
> **Failure example (1)**
>
> ## Failure examples
> ### Failure Examples in Chain of Thought that are resolved by SG
>
>
> 1. Context: A horse is in front of a dog from the dog's perspective
>
> CoT: Explanation: front relation is based on the dog's perspective to the horse even if dog has the intrinsic direction and a horse is not inside the dog. Answer: external relative.
>
> SG: Explanation: Topological: horse is not inside of the dog. Distance: horse may has some distance from dog according to the context. Direction: Dog which is the relative object has the direction, and horse is in front of the dog. Answer: external intrinsic.
>
> label: external intrinsic
>
> 2. Context: A cow is in front of a container and outside of the container
>
> CoT: Explanation: front relation is based on the container's intrinsic direction and the cow is not inside the container. Answer: external intrinsic.
>
> SG: Explanation: Topological: cow is outside of the container. Distance: cow may be little bit far from the front of the container from the context. Direction: Container which is the relative object doesn't have the direction, but the context front relation is referred from observer's perspective that cow is in front of the container location. Answer: external relative. However, it could also be interpreted as external intrinsic if we consider the container having a front direction. Without more specific information, the safer categorization would be external relative.
>
> label: external relative
>
> 3. Context: A fire hydrant is inside and on the left of a bus from the bus itself
>
> CoT: Explanation: The phrase "from the bus itself" implies that the reference frame is the bus, and the relation "inside" and "on the left" are described from the bus's perspective. Answer: internal relative.
>
> SG: Explanation: Topological: fire hydrant is inside of the bus. Distance: fire hydrant should be in the same location with the bus from the context. Direction: The bus which is the relative object has the direction, and fire hydrant is on the left side of the bus from the bus's own perspective. Answer: internal intrinsic.
>
> label: internal intrinsic
>
> 4. Context: A chicken is in front of a fire hydrant
>
> CoT: Explanation: position of the chicken is referring to the front direction of the fire hydrant, which is an intrinsic property of the fire hydrant. Answer: external intrinsic.
>
> SG: Explanation: Topological: chicken is outside of the fire hydrant. Distance: chicken may be close to the fire hydrant from the context. Direction: fire hydrant which is the relative object has no direction, and the context "in front of" relation is described from an observer's perspective. Answer: external relative.
>
> label: external relative
>
> 5. Context: A chair is back of a tree
>
> CoT: Explanation: position of the chair is referring to the back direction of the tree, and both objects are not containing one another. Answer: external intrinsic.
>
> SG: Explanation: Topological: chair is not inside of the tree. Distance: chair may have some distance from the tree according to the context. Direction: The tree does not have an intrinsic front or back, so the context is likely referring to the observer's perspective that the chair is behind the tree. Answer: external relative
>
> label: external relative
>
> 6. Context: A suitcase is on the right of a water tank
>
> CoT: Explanation: right relation is based on the intrinsic direction of the water tank and both objects are not containing one another. Answer: external intrinsic.
>
> SG: Explanation: Topological: The suitcase is not inside the water tank. Distance: The suitcase is at some distance from the water tank. Direction: The water tank does not have an intrinsic direction, so the right relation is inferred from the observer's perspective. Answer: external relative
>
> label: external relative
>
> 7. Context: A bus is in front of a fire hydrant
>
> CoT: Explanation: position of the bus is referring to the front direction of the fire hydrant, and both objects are not containing one another. Answer: external intrinsic.
>
> SG: Explanation: Topological: bus is not inside of the fire hydrant. Distance: bus may have some distance from the fire hydrant according to the context. Direction: The fire hydrant does not have an intrinsic front, so the context is referring to the bus's position relative to the fire hydrant from the observer's perspective. Answer: external relative
>
> label: external relative
>
> 8. Context: A cow is back of a bag
>
> CoT: Explanation: position of the cow is referring to the back direction of the bag, and both objects are not containing one another. Answer: external intrinsic.
>
> SG: Explanation: Topological: cow is not inside of the bag. Distance: cow may have some distance from the bag according to the context. Direction: The bag, which is the relative object, does not have an intrinsic direction, so the context is likely referring to the observer's perspective that the cow is behind the bag. Answer: external relative
>
> label: external relative

---

> ### Author Response · Authors · 2024-11-24
> **Failure example (2)**
>
> ### Failure Examples of SG-prompting
>
> 9. Context: A horse is in front of a fire hydrant
>
> SG: Explanation: Topological: horse is not inside of the fire hydrant. Distance: horse may has some distance from fire hydrant according to the context. Direction: Fire hydrant doesn't have the direction and context refere from observer perspective for horse is in front of the fire hydrant. However, since there's no mention of the observer's perspective, it falls under external intrinsic where the direction is inferred based on the intrinsic direction of the fire hydrant. Answer: external intrinsic.
>
> label: external relative
>
> 10. Context: An umbrella is in front of a container and outside of the container
>
> SG: Explanation: Topological: umbrella is not inside of the container. Distance: umbrella may has some distance from container according to the context. Direction: Container doesn't have the direction and context front relation is referred from observer's perspective that an umbrella is in front of the container location. However, since the umbrella is outside of the container, it falls under the external intrinsic category as it describes the relative position of the umbrella using the container's intrinsic direction. Answer: external intrinsic.
>
> label: external relative
>
> 11. Context: A water tank is in front of a chicken from the chicken itself
>
> SG: Explanation: Topological: water tank is not inside of the chicken. Distance: water tank may has some distance from chicken according to the context. Direction: the context "from the chicken itself" means the direction is referred from the chicken's perspective, and the water tank is in front of the chicken. Answer: internal relative.
>
> label: external intrinsic
>
> 12. Context: A container is in front of a deer from the deer itself
>
> SG: Explanation: Topological: container is not inside of the deer. Distance: container may has some distance from deer according to the context. Direction: the context "from the deer itself" means the deer is the observer, and the direction is referred from the deer's perspective that a container is in front of the deer. Answer: internal relative.
>
> label: external intrinsic
>
> 13. Context: A box is back of an umbrella
>
> SG: Explanation: Topological: box is not inside of the umbrella. Distance: box may has some distance from umbrella according to the context. Direction: The umbrella which is the relative object has the direction and box is actually in back of the umbrella. Answer: external intrinsic
>
> label: external relative
>
> 14. Context: A chicken is behind a fire hydrant
>
> SG: Explanation: Topological: chicken is not inside of the fire hydrant. Distance: chicken may have some distance from the fire hydrant according to the context. Direction: The fire hydrant, which is the relative object, has an intrinsic front and back, and the chicken is behind it. Answer: external intrinsic
>
> label: external relative
>
> 15. Context: A cow is in front of a container and outside of a container
>
> SG: Explanation: Topological: cow is outside of the container. Distance: cow may have some distance from the container according to the context. Direction: The container, which is the relative object, has an intrinsic front side, and the cow is positioned in front of it. Answer: external intrinsic
>
> label: external relative
>
> 16. Context: An umbrella is on the back of a fire hydrant
>
> SG: Explanation: Topological: umbrella is not inside of the fire hydrant. Distance: umbrella may have some distance from the fire hydrant according to the context. Direction: The fire hydrant, which is the relative object, has an intrinsic front and back, and the umbrella is positioned at the back of the fire hydrant. Answer: external intrinsic
>
> label: external relative

---

> ### Author Response · Authors · 2024-11-24
> **Response (2)**
>
> ## Lack of full prompts for different settings
>
> We apologize for not including the full prompt in the main text due to lack of space.
> Appendix C provides all prompt settings and some explanations of each setting.
> We will emphasize this more in the revised version.
>
> ## Setting for SG-prompting
>
> The SG prompting follows a few-shot setting to help LLMs recognize the response patterns associated with various spatial relations (directional, topology, and distance). Sorry for the confusion. We refer to examples in the appendix in line 290. Some examples are included in Appendix C, and we intend to incorporate additional examples into the revised version of the main paper.
>
> ## Abnormal results of GPT family
>
> As shown in Figure 4, the GPT4 family mostly responds that the spatial relation originates from the relatum (intrinsic FoR class). The outperformance of GPT-3.5 is due to its bias to External relative and the fact that many ambiguous cases in A-split can fall in the external relative. We explained this in Lines 385-387 for the result of Gemma2-9B, but we will make sure this is well highlighted in the new version.
>
> Overall, the GPT family, epically in the GPT-4o, is biased towards assuming intrinsic direction for each object, leading to lower results in both the A-split and the EI/II of the C-split.
> We also manually verified this based on the responses from Chain-of-Thought and SG-prompting, which indicates that the models occasionally claim that objects without intrinsic direction, such as trees, possess a front direction while at other times stating otherwise. We clarify this using an example in the appendix.
>
>
> **We hope that all responses address any confusion the reviewer raised. We hope they are convincing about our work and could increase our score. Again, we really appreciate your comment. We will let you know again when we upload the revised version.**

---

> > ### Comment · Reviewer_WWK9 · 2024-11-25
> >
> > Thank you for explanation. All of my previous questions are resolved, while weaknesses still remain unaddressed.
> >
> > > 1. The dataset is pure synthetic and constructed by a limited number of textual templates.
> >
> > "Though this simplifies reasoning if the language model can capture the pattern, our experiments show that it can still fail to recognize the frame of reference." Do you mean failure cases indicates LLM's spatial limitation? I can't agree with that since this task fails to disentangle linguistic and spatial ability of LLMs.
> > - As pointed out by reviewer xDEW, the definition of the terms provided to the LLMs is formal and not intuitive, which might contribute to the classification error.
> > - In most of the cases, at least for humans, the expression can be solved by rules, without really understanding the spatial configuration of the scene.
> >
> > > 2. Inductive bias in textual templates might be leveraged by SG-prompting.
> >
> > This is verified when comparing the **SG (no template)** and **SG (with template)** column in the table provided. Moreover,  **CoT  (no template)** and **CoT (with template)** also shows the bias introduced by textual templates.
> >
> > > 3. full prompts of different settings
> >
> > There's a misunderstanding in the prompts and demonstrations. I thought "example responses" in listing 2 in Appendix C are output showcases for different settings. But it turns out to be the few-shot demonstrations. I think **the explanations in CoT setting are unfair by using shortened descriptions, ambiguous terms and typos. It hinders the rigorousness of this paper.**
> > - line 834: "The bird is accuracy left of the car. Answer: internal intrinsic."
> > - line 844: "which makes the back relation based on the observer’s perspective of the room"
> >
> > Based on above, I'll decrease my score.

---

> ### Author Response · Authors · 2024-11-25
>
> **Prompt Quality of COT**
> We are sorry that you are more critical about the quality of our prompt for the COT case and made you reduce the overall assessment of our work.  Here we would like to provide our further defense about this situation.
>
>  - Based the comment that we received from reviewer  xDEW, we changed the prompt and the re-ran the experiments,  as expected the results are sensitive to small changes in the language, so, we obtain some different values for some of the settings, however, the new results still indicate the outperformance of our SG-prompting compared to COT. We think the advantage comes from characterizing the type of spatial relations  in SG and this is consistently observed independent from the quality of  prompt's phrasing.
>
> - To further confirm the validity of our conclusions, in obtaining the new results on Qwen in the response to reviewer bEBN, we did extra caution  about the quality of phrasing the prompt and typos and used a simpler phrasing.  As you see the results are consistent with out previous conclusions.
>
>  - The length of COT explanations is comparable to the length of SG explanation. In the worst case, there is two sentences difference.
>
>  - A minor justification of some of the typos also is that the example prompts in the appendix included a typo that was not in the actual prompt "The bird is accuracy left of the car" was actually "The bird is accurately left of the car" but it seems, in the appendix, that spelling typo happened.
>
> We hope these new pieces of information and our new reported results are convincing for the reviewer to change their mind and do not relate the merits of the paper to the typos in prompt.

---

> ### Author Response · Authors · 2024-11-25
>
> **Bias caused by templates**
>
> We believe there is a misunderstanding regarding the advantage of using explicit templates in a portion of our dataset. The purpose of these templates is to disambiguate the Frame of Reference (FoR) in linguistic expressions. We do not think there is significant variety in linguistic utterances for specifying perspective since they typically rely on the relatum, observer, or the speaker's point of view, all of which are addressed in our templates. Therefore, we argue that our templates cover the various ways perspective can be expressed in language. Moreover, we demonstrate that explicitly expressed perspectives help language models more easily recognize the FoR class.
> It is important to note that this applies to **only one split of our dataset. We also include other splits where the FoR is implicit.** In cases where the FoR is not explicitly mentioned in the text, we still aim for the models to recognize all possible valid FoRs. This explains the observed bias, which we do not consider a disadvantage. Instead, we hypothesize that explicit perspective information benefits the model, and we seek to leverage this further through improved prompting techniques.
> Therefore in our approach, we instruct the model to recognize the FoR based on object affordance (e.g., container vs. non-container, possessing intrinsic direction vs. not possessing it) and the types of relationships proposed in the SG prompting framework. We refer to the new table of Qwen as evidence (in addition to all results in our paper) that SG prompting improves accuracy by 26% for non-templated cases compared to 5.5% for templated cases. This difference arises because the templated cases were already straightforward; however, even for these cases, encouraging the model to focus on the type of relationship still provided improvements.
>
> Finally, please refer to the examples in our previous response, which illustrate the advantage of SG prompting for untemplated text.
>
> **We hope our contribution for both dataset synthesis, evaluation and the prompting solution is more clear here and the reviewer is more convinced about the merits and quality of our works.**

---

> ### Author Response · Authors · 2024-11-26
>
> Based on the reviewer's comment regarding the short explanation in the CoT, we did additional experiments, see below Table. In this experiment, we extend the CoT examples to include more information regarding the direction of relatum to identify the FoR classes. We provide the result on Qwen2 with the new CoT explained above. According to the table, we observe some changes in results. The results from the new CoT show the model favors their preference class (Relative class for Qwen2). The preference of Qwen2 is identified based on our additional results [see Tables in response to reviewer bEBN]. Overall, the old CoT prompt provides a better average compared to the new COT, so this change does not influence the main conclusion of our experiments.
>
> | Model  | ER | EI | II | IR | All |
> |----------|--------|--------|--------|--------|--------|
> | New CoT | 93.39 | 67.72 | 79.69 | 87.10 | 81.97 |
> | Old CoT | 84.69 | 78.26 | 92.19 | 85.89 | 85.26 |
>
> **We also want to remind the reviewer of other contributions to our paper besides the SG prompting. We present the FoREST dataset to reveal the LLMs' understanding of the frame of reference, which is important for comprehending spatial language. Most current spatial benchmarks pay less attention to this aspect and assume the same frame of reference across all scenarios. Our results reveal that different LLMs interpret spatial expressions differently, which could influence the model's performance in more complex tasks.
> We also provide the results of a text-to-image task, which confirm our hypothesis that information regarding FoR potentially enhances the performance of the downstream task.
> We hope this is convincing enough for the reviewer to reconsider our paper's value and overall assessment.**

---

### Official Review · Reviewer_m8vc · 2024-11-02

**Soundness:** 3
**Presentation:** 3
**Contribution:** 4
**Rating:** 6
**Confidence:** 3

**Summary:**

The paper presents the FoREST benchmark, aimed at testing large language models' (LLMs) understanding of frames of reference (FoR) in spatial reasoning. FoR refers to different perspectives (intrinsic, relative, and absolute) used to describe spatial relationships. The benchmark assesses LLMs' ability to identify FoR in ambiguous and clear spatial contexts and perform text-to-image generation accordingly. Results show that LLMs have biases in FoR interpretation, impacting spatial layout accuracy. They also introduce Spatial-Guided prompting to improve FoR comprehension and performance in related tasks.

**Strengths:**

- **Novel Perspective**: Introduces an innovative approach to assessing spatial perception in large models, focusing on frames of reference (FoR).
- **Theoretical Support**: Draws on established spatial language literature to support the motivations and foundational concepts of FoR.
- **Insightful Analysis**: Offers valuable insights into both FoR identification and text-to-image mapping.

**Weaknesses:**

No dataset or code provided

**Questions:**

Could you make the datasets and code available?

---

> ### Author Response · Authors · 2024-11-21
>
> We highly appreciate the reviewer's recognition of our novel contribution and the theoretical support we provide in handling frames of reference in spatial language.
> We have already attached the supplementary material, including the dataset's textual part; we will publish the code and the visual part when the non-anonymized paper is made available to the public.
> We are also working on our anonymous GitHub for the reviewer period. We will update it when we have the link.
> Again, we really appreciate your comment.
> Please feel free to share any concerns you may have that we can address to improve our rating.

---

### Official Review · Reviewer_xDEW · 2024-11-03

**Soundness:** 2
**Presentation:** 3
**Contribution:** 2
**Rating:** 5
**Confidence:** 3

**Summary:**

This paper proposes a new Frame of Reference (FoR) comprehension task for LLMs, where the models need to identify the perspective category based on a given spatial narrative. For this task, a new benchmark, named FoREST is generated. Using this benchmark, the paper identifies the inability and biases of various LLMs in solving this task, and proposes a new prompting technique to guide LLMs in identifying key information from the textual input thus enhancing their performance on this task. This paper also shows how this ability can be utilized for text-to-image generation under specific spatial guidance, highlighting the potential application value of this work.

**Strengths:**

The paper proposes a novel task that can potentially reveal the ability of LLMs in understanding spatial concepts.

**Weaknesses:**

1. It is unclear if the proposed Spatial-Guided prompting technique helps “reduce FoR bias in LLMs”, as claimed in the abstract in the paper, or just clarify the category terms that LLMs are tasked to identify. Since the FoR classes (external intrinsic, external relative, etc) are technical terms in cognitive studies that do not appear commonly in the internet data used for training LLMs, a clear and intuitive explanation of the terms is naturally important for solving this task. However the definition of the terms provided to the LLMs is formal and not intuitive. For example, it does not define clearly what does “the referenced object’s intrinsic directions” mean. This is only explained to some extent in the Spatial-Guided prompting examples, such as “the car has direction”. What if the concepts are explained in plainer and more intuitive language? Such as:

$\textit{"External intrinsic: The spatial description of an item A relative to another item B, where (1) A is not contained by B; (2) The spatial relationship description is relative to the B’s facing direction, if B has one (Example: a horse, a car. Counterexample: a box.)"}$

2. Why understanding FoR is an important problem is not articulated adequately. Since, according to the introduction in this paper, this task is more commonly seen cognitive linguistics study rather than AI or related fields, more discussion on the potential application of FoR understanding ability of LLMs can help readers better understand the motivation. The text-to-image task shown in the paper is a great application, but it is on specifically designed command types. Can this ability be potentially applied for other embodied AI or robotic tasks that require strong spatial understanding capacity?

**Questions:**

Does the temperature setting impact the bias of LLMs on this task? The paper sets the sampling temperature to 0, and claims the bias of LLMs by showing that they more frequently produce external classes under ambiguous queries (that correspond to multiple correct categories) in Figure 4. It is possible that a low temperature limits the diversity of LLMs response.

---

> ### Author Response · Authors · 2024-11-21
>
> We appreciate your valuable feedback on our work.
> We would like to address each weakness you mentioned and the question below.
>
> ## Weaknesses
> ### Prompt is not well-explain
>
> We agree that our prompting can be rephrased and simplified. We then experimented with your suggested phrasing.
> We report the results in the Table below based on your prompt. You can compare this with Table  1 in the paper.
> As you can see, the results are very sensitive to changes in the prompt and vary for either better or worse. However, our prompt provides a better average, so this change does not influence the main conclusion of our experiments.
>
> | Model              | ER              | EI              | II               | IR               | Avg              |
> |--------------------|-----------------|-----------------|------------------|------------------|------------------|
> | Llama3-8B (0-shot) | 48.63           | 94.02           | 78.91            | 6.45             | 57.00            |
> | Llama3-8B (4-shot) | 53.93 ↑5.30     | 56.85 ↓37.17    | 100.00 ↑21.09    | 37.90 ↑31.45     | 62.17 ↑5.17      |
> | Llama3-8B (CoT)    | 63.55 ↑14.92    | 42.28 ↓51.74    | 93.75 ↑14.84     | 35.48 ↑29.03     | 58.77 ↑1.76      |
> | Llama3-8B (SG)     | 69.31 ↑20.68    | 79.02 ↓15.00    | 100.00 ↑21.09    | 19.76 ↑13.31     | 67.02 ↑10.02     |
>
> *Table: Results of C-split of Llama3-8B with updated prompt. "↑" indicates improvement over 0-shot, and "↓" indicates decrease compared to 0-shot.*
>
> ### Changing the bias of SG prompting
>
> We apologize for the confusion of the purpose of SG prompting.
> Our primary intention is not to change the bias as long as the model has a correct interpretation.
> Initially, the model preferred some specific FoR that was possibly incorrect.
> Therefore, we want to direct the model to describe related spatial relations and provide better responses, which could potentially change the inherent bias of the model towards specific classes.
> This, in turn, improves FoR comprehension and performance in related tasks.
> We will update our abstract to emphasize improving accuracy instead of a change in their bias.
>
> ### Lack of discussion regrading the potential application used FoR understanding
>
> We agree that adding the potential applications in the introduction would enhance the reader’s understanding of the motivation behind characterizing the FoR.
> Currently, we discuss how current AI benchmarks and the lack of utilization of the FoR in their dataset from lines 42 to 50.
> We will rephrase this part to include more information on the potential problems in these applications that necessitate a comprehensive understanding of the FoR.
> Certainly, embodied AI is one important application that will benefit from FOR comprehension, particularly when an instruction-giver and instruction-follower have different perspectives and potential variations in their spatial language and usage of FORs.
> This requires the model to comprehend the dynamic change in FoR (perspective changes) in the instruction so that it can perform the task more effectively.
> Other potential applications, such as video narrative generation and 3D scene construction based on text, can also benefit from this ability since they require the model to understand different perspectives.
> ***We are going to integrate these explanations and motivation in the new version of the paper.***

---

> > ### Comment · Reviewer_xDEW · 2024-11-22
> > **Reviewer Response**
> >
> > I would like to thank the authors for their detailed explanation and the additional experimental results. As a follow-up, could you provide further analysis or qualitative demonstrations of baseline failure cases, particularly for CoT? I remain concerned about whether the proposed SG prompting genuinely enhances LLMs’ spatial understanding or if it primarily leverages linguistic cues to categorize sentences with only superficial spatial interpretation, given that the inputs are generated from a limited set of templates. A detailed analysis of LLM outputs might help address this concern.

---

> ### Author Response · Authors · 2024-11-21
>
> ## Questions
>
> **Does the temperature setting impact the bias of LLMs on this task?**
>
> It is possible that temperature influences the bias of LLMs, particularly in the zero-shot setting.
> To address this, we conducted experiments with Llama3-70B.
> Comparing two distinct temperatures (0 and 1) revealed a change in the distribution, that is the frequencies of the classes changed sometimes for 10%. However, the change is not dramatic, and it seems the relative preferences for most of the categories did not change.
> Specifically, the model showed the same highest frequency responses for the cow, car, and pen cases, even higher frequency in some settings.
> Therefore, a high temperature does not significantly change the diversity of LLMs' responses to this task, which is an interesting result. We is going to add the related tables to the appendix of the new version due to the lack of space.
>
>
> ### Cow Case
>
> | Model   | ER temp-0 | ER temp-1 | EI temp-0 | EI temp-1 | II temp-0 | II temp-1 | IR temp-0 | IR temp-1 |
> |---------|-----------|-----------|-----------|-----------|-----------|-----------|-----------|-----------|
> | 0-shot  | 75.38     | 87.12     | 23.86     | 12.50     | 0.76      | 0.13      | 0.00      | 0.25      |
> | 4-shot  | 0.00      | 15.66     | 100.00    | 84.34     | 0.00      | 0.00      | 0.00      | 0.00      |
> | CoT     | 31.82     | 49.87     | 68.18     | 49.87     | 0.00      | 0.13      | 0.00      | 0.13      |
> | SG      | 51.39     | 70.45     | 48.61     | 29.42     | 0.00      | 0.00      | 0.00      | 0.13      |
>
> ### Box Case
>
> | Model   | ER temp-0 | ER temp-1 | EI temp-0 | EI temp-1 | II temp-0 | II temp-1 | IR temp-0 | IR temp-1 |
> |---------|-----------|-----------|-----------|-----------|-----------|-----------|-----------|-----------|
> | 0-shot  | 22.50     | 41.67     | 77.50     | 58.33     | 0.00      | 0.13      | 0.00      | 0.25      |
> | 4-shot  | 0.00      | 0.00      | 100.00    | 100.00    | 0.00      | 0.00      | 0.00      | 0.00      |
> | CoT     | 0.00      | 5.83      | 100.00    | 94.17     | 0.00      | 0.00      | 0.00      | 0.00      |
> | SG      | 11.67     | 33.33     | 88.33     | 66.67     | 0.00      | 0.00      | 0.00      | 0.00      |
>
> ### Car Case
>
> | Model   | ER temp-0 | ER temp-1 | EI temp-0 | EI temp-1 | II temp-0 | II temp-1 | IR temp-0 | IR temp-1 |
> |---------|-----------|-----------|-----------|-----------|-----------|-----------|-----------|-----------|
> | 0-shot  | 55.20     | 68.24     | 49.01     | 31.15     | 0.79      | 0.61      | 0.00      | 0.00      |
> | 4-shot  | 0.60      | 5.94      | 99.40     | 94.06     | 0.00      | 0.00      | 0.00      | 0.00      |
> | CoT     | 19.64     | 38.52     | 80.16     | 61.27     | 0.20      | 0.20      | 0.00      | 0.00      |
> | SG      | 44.25     | 56.97     | 55.75     | 43.03     | 0.00      | 0.00      | 0.00      | 0.00      |
>
> ### Pen Case
>
> | Model   | ER temp-0 | ER temp-1 | EI temp-0 | EI temp-1 | II temp-0 | II temp-1 | IR temp-0 | IR temp-1 |
> |---------|-----------|-----------|-----------|-----------|-----------|-----------|-----------|-----------|
> | 0-shot  | 90.62     | 96.88     | 9.38      | 3.12      | 0.00      | 0.61      | 0.00      | 0.00      |
> | 4-shot  | 0.00      | 7.03      | 100.00    | 92.97     | 0.00      | 0.00      | 0.00      | 0.00      |
> | CoT     | 17.19     | 28.91     | 82.81     | 71.09     | 0.20      | 0.20      | 0.00      | 0.00      |
> | SG      | 48.31     | 57.81     | 54.69     | 42.19     | 0.00      | 0.00      | 0.00      | 0.00      |
>
> *Table: The results between two different temperatures of Llam3-70B on the A-split of FoREST. The numbers show the percentage frequency of responses from the model.*
>
> We hope that all responses address any confusion the reviewer raised. We hope they are convincing about our work and could increase our score. Again, we really appreciate your comment. We will let you know again when we upload the revised version.

---

> ### Author Response · Authors · 2024-11-24
>
> Thank you for the quick follow-up. We provide the additional results from the C-split, which context has topology/perspective templates for clarifying the FoR ambiguity in the language.
> Again, these templates are added to create a split dataset with unambiguous FoR classes, given the language's possible ambiguity in FoR.
> This information in the template helps the language model to use this information for spatial inference/reasoning.
>
> Our results demonstrate that SG Prompting significantly enhances the FoR classification accuracy, particularly in the context without such a template. This is evident in the table below and the qualitative example provided.
> Furthermore, the results from CoT suggest that the models prefer to categorize the context using linguistic cues rather than considering other spatial relations that we explicitly guide the model to consider during SG prompting.
> So, our SG-prompting enhances LLMs' spatial understanding rather than relying too much on linguistic clues.
> We also provide some qualitative examples of failure cases in response to the reviewer WWK9 (failure example (1) and failure example (2))
>
> | Model      | CoT (no template) | SG (no template) | CoT (with template) | SG (with template) |
> |------------|-------------------|------------------|---------------------|--------------------|
> | Gemma-9B   | 2.58              | 35.51 (↑ 32.93)  | 72.65               | 73.80 (↑ 1.15)     |
> | Llama3-8B  | 22.22             | 36.90 (↑ 14.68)  | 73.64               | 71.07 (↓ 2.57)     |
> | Llama3-70B | 19.84             | 44.64 (↑ 24.80)  | 76.72               | 87.39 (↑ 10.67)    |
> | Qwen2      | 58.20             | 84.22 (↑ 26.02)  | 88.36               | 93.86 (↑ 5.50)     |
> | GPT-3.5    | 1.58              | 43.25 (↑ 41.67)  | 77.64               | 85.21 (↑ 7.57)     |
> | GPT-4o      | 12.50             | 29.17 (↑ 16.67)  | 87.73               | 90.74 (↑ 3.01)     |
>
> *Table: The table presents the results of our C-Split experiment on our dataset. The “↑” symbol indicates an improvement over the CoT baseline, while the “↓” symbol denotes a decrease compared to the CoT baseline. The table is divided into two sections: context with templates and context without templates. It is important to note that the context without templates is inherently clear and does not require additional information.*
>
> **We will include table in the main paper and examples in the appendix for the qualitative analysis of our experiment**

---

> > ### Comment · Reviewer_xDEW · 2024-11-29
> >
> > Thanks to the authors to providing more experiment results and failure examples. These answer some of my questions but other main concerns remain. It is still not clear to me if the proposed task really evaluates on the spatial understanding ability of LLMs or just linguistic analysis in this specific context. Furthermore, it looks like one of the major reasons for the failure of CoT is the inability to determine what is "intrinsic direction", which seems to be a subjective concept that is not well defined in the prompt. Example:
> > >Context: A cow is in front of a container and outside of the container
> > >
> > > CoT: Explanation: front relation is based on the container's intrinsic direction and the cow is not inside the container. Answer: external intrinsic.
> >
> > In this case, even SG is not certain about this concept, although it makes a correct guess.
> > > SG: Explanation: Topological: cow is outside of the container. Distance: cow may be little bit far from the front of the container from the context. Direction: Container which is the relative object doesn't have the direction, but the context front relation is referred from observer's perspective that cow is in front of the container location. Answer: external relative. **However, it could also be interpreted as external intrinsic if we consider the container having a front direction.** Without more specific information, the safer categorization would be external relative.
> > >
> > > label: external relative
> >
> > Nonetheless, this paper introduces a novel spatial task, shows bias in LLMs, and highlights the importance of prompt structure in aiding LLMs' comprehension of the task. While I would consider raising the score, my overall stance remains negative regarding the acceptance of the work.

---

> > > ### Author Response · Authors · 2024-11-29
> > >
> > > > Spatial understanding
> > >
> > >
> > > Our claim is that identifying FoR (indeed based on linguistic analysis) helps better spatial understanding as measured in the downstream tasks.
> > > Our approach is to analyze the spatial language better and provide the LLMs with explicit knowledge of the FoR so that they can analyze the linguistic surface and the properties of landmarks to infer the FoR and use it for spatial reasoning later.
> > > Spatial reasoning can be manifested in tasks such as T2I.
> > > In T2I, placing objects in the correct relative location indicates a better spatial understanding.
> > > We show that the linguistic analysis and the knowledge of FoR extracted from SG prompting provide LLMs with a better spatial understanding when creating the spatial layout for the T2I model.
> > >
> > > > SG prompting
> > >
> > > We want to clarify the confusion about our proposed SG approach and CoT approach. We would want to remind you that our proposed prompting approach can also be called CoT since we only try to specify the importance concept for reasoning to identify FoR. The main question is how to explain the reasoning steps for identifying FoR. The innovation of our approach is to characterize the important spatial concepts involved to help improve FoR comprehension. We try to have the baseline CoT that does not focus on these explicit concepts, and we want to show that including these concepts (type of spatial relations, topology, distance, and direction in addition to properties of the relatum) in SG promoting (variation of CoT) helps model for better FoR identification.

---

### Author Response · Authors · 2024-11-29
**General comment related to SG vs CoT**

We want to clarify the confusion about our proposed SG approach and CoT approach.
We would want to remind you that our proposed prompting approach can also be called CoT since we only try to specify the importance concept for reasoning to identify FoR.
The main question is how to explain the reasoning steps for identifying FoR.
The innovation of our approach is to characterize the important spatial concepts involved to help improve FoR comprehension.
We try to have the baseline CoT that does not focus on these explicit concepts, and we want to show that including these concepts (type of spatial relations, topology, distance, and direction in addition to properties of the relatum) in SG promoting (variation of CoT) helps model for better FoR identification.

---

> ### Author Response · Authors · 2024-12-02
>
> We would also like to emphasize that, in addition to proposing SG prompting to enhance frame of reference identification, the primary contribution of this paper is the evaluation framework we introduce to assess LLMs' understanding of spatial frames of reference. We demonstrate the significance of defining the FoR concept in spatial reasoning and layout generation by LLMs for downstream tasks, such as text-to-image generation. We sincerely appreciate the reviewers' time and dedication in providing detailed feedback, and we hope they will consider the multifaceted contributions of our research presented in this paper.

---

### Meta-Review · Area_Chair_V5Uf · 2024-12-18

**Metareview:**

The paper introduces a frame of reference (FoR) comprehension task for LLMs. The paper specifically focuses on understanding the perspective of spatial relations. For example, can LLMs distinguish between "a cat is to the right of the car from the car's perspective" vs "a cat is to the left of a car from my perspective". The task is posed as a multi-class classification task. The authors test different prompting strategies for this task, and find that textual descriptions of topological relations (inside vs not), frame of reference and distances (far) can help the models perform better on the task.

Understanding spatial relationships is an important skill and the paper further LLM's limited understanding of spatial relationships. Using text-to-image for understanding spatial understanding of LLMs

The reviewers actively engaged with the authors during rebuttal, but several of the reviewer's concern remained. Specifically, reviewers are concerned that in this instantiation of the task, it is unclear if it tests the spatial understanding of LLMs or just linguistic understanding. Additionally, reviewers believe that further investigation and prompt engineering is required to make the experiments more sound.

**Additional Comments On Reviewer Discussion:**

The reviewers actively engaged with the authors during rebuttal. After the discussion, reviewers were still tending negative and felt several of their concerns remained unaddressed:

The reviewers remain concerned that the the task setup tests the LLM's spatial understanding or not, specially because the inputs are generated from a limited set of templates (xDEW, WWK9). The reviewers also felt that the lack of templates in the dataset construction also makes it harder to truly understand the LLMs capability.  Reviewer bEBN also raised concerns about what insights can be drawn from the benchmark to improve LLMs spatial understanding. Paper will benefit from more careful experimental design, and going beyond prompt engineering to fix LLMs bias for spatial understanding.

---

### Decision · Program_Chairs · 2025-01-22

Reject